# Reconfigurable signal modulation in a ferroelectric tunnel field-effect transistor

Zhongyunshen Zhu [1] ✉, Anton E. O. Persson [1] & Lars-Erik Wernersson [1] ✉

Reconfigurable transistors are an emerging device technology adding new functionalities while lowering the circuit architecture complexity. However, most investigations focus on digital applications. Here, we demonstrate a single vertical nanowire ferroelectric tunnel field-effect transistor (ferro-TFET) that can modulate an input signal with diverse modes including signal transmission, phase shift, frequency doubling, and mixing with significant suppression of undesired harmonics for reconfigurable analogue applications. We realize this by a heterostructure design in which a gate/source overlapped channel enables nearly perfect parabolic transfer characteristics with robust negative transconductance. By using a ferroelectric gate oxide, our ferro-TFET is non-volatilely reconfigurable, enabling various modes of signal modulation. The ferro-TFET shows merits of reconfigurability, reduced footprint, and low supply voltage for signal modulation. This work provides the possibility for monolithic integration of both steep-slope TFETs and reconfigurable ferro-TFETs towards high-density, energy-efficient, and multifunctional digital/analogue hybrid circuits.

With the transistor size scaling approaching its physical limit, higher functional density is becoming an increasingly desirable technological complement[1]. Reconfigurable field-effect transistors (FETs) are such an attractive class of devices offering more functionality for the same number of devices[2]. Unlike conventional metal-oxide-semiconductor FETs (MOSFETs) in which the polarity (n- and p-type) and threshold voltage ($V_T$) is predefined, reconfigurable FETs can change these properties after manufacturing. A large family of reconfigurable FETs are the Schottky barrier transistors[3,4] where the polarity is controlled by the height of a Schottky barrier through an extra gate. Another group, ferroelectric FETs (FeFETs), which replace the dielectric oxide with a ferroelectric oxide in otherwise conventional MOSFETs, can shift the $V_T$ by tuning the ferroelectric polarization with the application of a voltage pulse to the gate electrode. FeFETs with tuneable $V_T$ have been demonstrated as reconfigurable logic gates[5] and run-time reconfigurable inverter-buffer logic[6]. However, most of the current reconfigurable FETs are designed for digital circuit applications such as multifunctional logic gates[5,7] and hardware security[4,6]. Extending transistor reconfigurability to analogue circuits for co-integration with

the logic modules on the same technology platform is of great importance for Internet-of-Thing (IoT) applications[8]. Recently, fundamental schemes of signal modulation used in communication systems, including frequency multiplication[9] and mixing[10] with reconfigurable operation based on FeFETs were demonstrated. The devices achieved ambipolar electrical transport by gate-induced drain leakage current and a reconfigurable $V_T$ shifted by the polarization in the ferroelectric gate. Additionally, a back-gate controlled reconfigurable Schottky FET has shown merged functionalities containing signal following, frequency doubling, and phase shifting[11]. Despite the reconfigurability achieved in these devices for analogue applications and full compatibility with Si complementary-metal-oxide-semiconductor (CMOS) technology, the supply voltage ($V_{DD}$) far above 1 V used to achieve the parabolic transfer characteristic raises the power consumption considerably, hence impeding their use in low-power communication systems such as IoT.

The tunnel FET (TFET) is a steep-slope device that can operate below the thermally limited subthreshold swing (SS) of 60 mV/decade by utilizing band-to-band tunnelling (BTBT) as the transport

[1]Department of Electrical and Information Technology, Lund University, 221 00 Lund, Sweden. ✉e-mail: zhongyunshen.zhu@eit.lth.se; lars-erik.wernersson@eit.lth.se

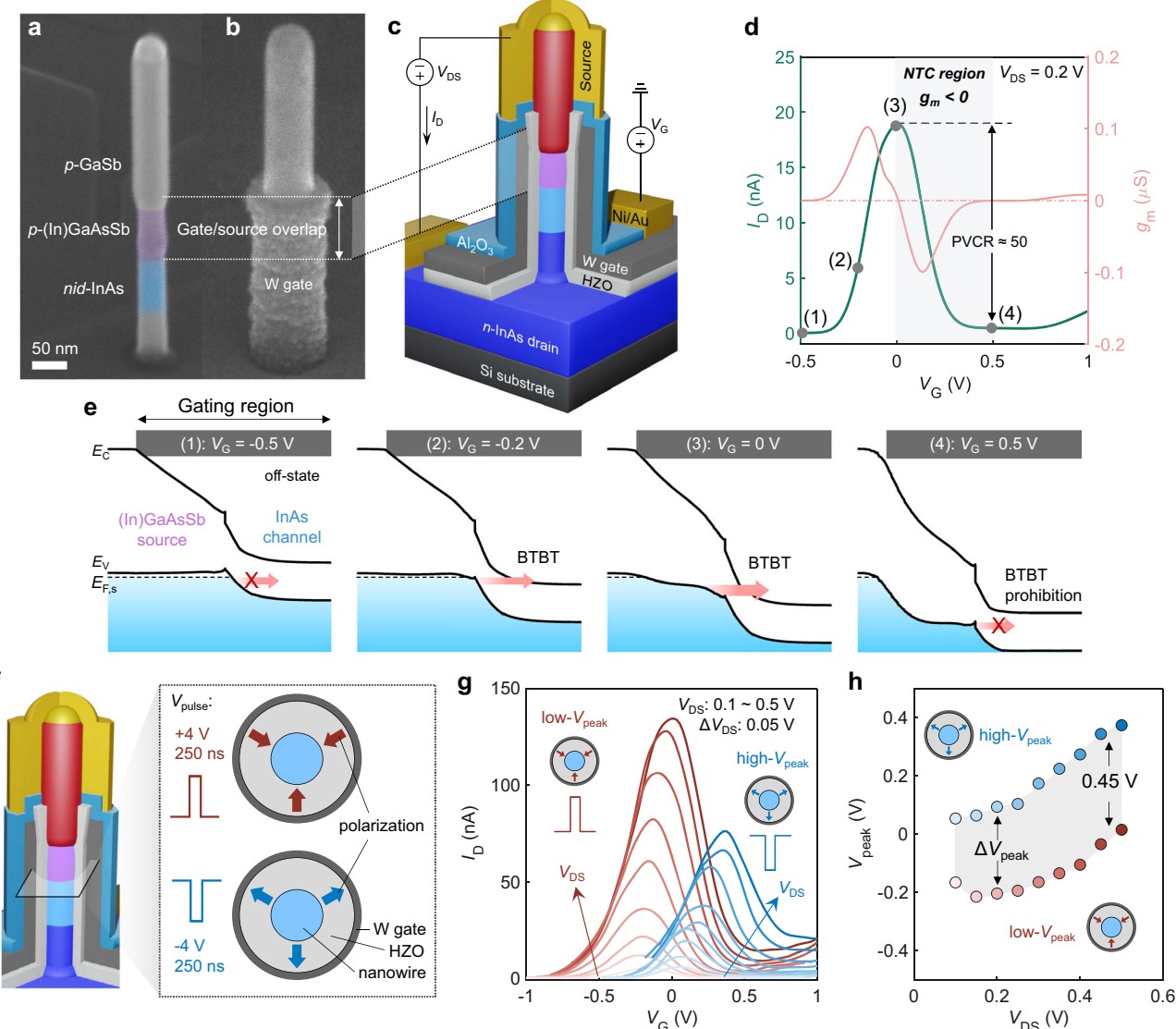

**Fig. 1 | Reconfigurable NTC in ferro-TFETs. a, b** SEM of (**a**) an as-grown nanowire heterostructure for a TFET and (**b**) a ferro-TFET post gate-length definition showing gate/source overlap. **c** Schematic of the final ferro-TFET device and the corresponding electrical measurement setup scheme. $V_G$, applied gate voltage; $V_{DS}$, source-drain bias; $I_D$, drain current. **d** Transfer characteristic with NTC realized by geometrical gate/source overlap. **e** Schematic band diagrams at different $V_G$ defined in (**d**), demonstrating prohibited BTBT when further increasing $V_G$. $E_C$, the conduction band energy; $E_V$, the valence band energy; $E_{F,s}$, the Fermi level of the source. **f** Cross-sectional schematic of the nanowire channel region at which the polarization of the ferroelectric gate oxide can be used to reconfigure the TFET by applying a voltage pulse of +4 V or −4 V for 250 ns at the gate, respectively. **g** Transfer characteristics of a ferro-TFET with two different polarizations. $V_{peak}$ (defined as $V_G$ at the $I_D$ peak) in the $I$–$V$ curves shows a positive shift when increasing $V_{DS}$ in both cases. Here, we define the two polarization states as low-$V_{peak}$ and high-$V_{peak}$ state, respectively, as displayed in the inset. **h** $V_{peak}$ as a function of $V_{DS}$. $\Delta V_{peak}$ is defined as the difference between two peak voltages and increases with $V_{DS}$.

mechanism, thereby being promising to lower the supply voltage[12,13]. Due to superior transport property and flexible material selection[14], III-V heterostructure TFETs have shown both high drain current ($I_D$) and SS well below 60 mV/decade at room temperature[15–18], which is imperative for low-power circuits. Besides, by tuning the gate alignment to overlap the source-channel tunnel junction in such heterostructure TFETs, a negative transconductance (NTC) can be achieved at high gate voltage ($V_G$)[18–20], creating a concave parabola-like transfer curve. Advantageously, NTC in III-V heterostructure TFETs remains and even becomes stronger at lower source-drain bias ($V_{DS}$)[18,19], allowing low-power analogue signal modulations such as phase shifting[21]. In the case of frequency doubling, compared to utilizing the current minimum in the symmetric transfer characteristic of ambipolar TFETs[22,23], NTC in TFETs allows frequency doubling around the maximum $I_D$ in the on-state, which increases the operating frequency. Therefore, transistors with NTC have been recently under tremendous interests and

intensively investigated in two-dimensional (2D) materials[24,25] and organic heterojunctions[26,27] with potentials for multi-valued logic gates[24,26], artificial synapses[25], and frequency doubling[28,29]. However, most of the demonstrated NTC-based transistors operate at high $V_G$ bias with a narrow NTC regime in the transfer characteristics due to the immature gate-stack development in these emerging materials. Therefore, further improvements are required to implement single NTC-based FETs for signal modulations.

In this article, we combine the reconfigurability and NTC-based signal modulation within a single ferroelectric TFET (ferro-TFET). Our approach relies on the successful integration of a ferroelectric Zr-doped HfO₂ (HZO) gate stack on a III-V nanowire gate-all-around TFET and a significant NTC achieved by designing a gate/source overlap structure in the ferro-TFET (Fig. 1a, b). This heterogeneous integration enables diverse functionalities of signal modulations including frequency transmission, doubling, mixing, and phase shift in a single

device, and these modes can be non-volatily reconfigured by the polarization of the ferroelectric gate. As a signal modulator, our ferro-TFETs excels in significantly reduced device footprint (~0.01 μm²) and supply voltage (down to 50 mV) while retaining high output power concentration at target frequency. The result benefits from the vertical nanowire structure and the almost ideal parabolic shape of transfer curve at various $V_{DD}$ which substantially suppresses harmonics without filters.

## Results

### Ferro-TFETs with reconfigurable NTC

The nanowire TFET structure consists of three main segments, $n$-type doped InAs as the drain, non-intentionally doped (*nid*) InAs as the channel, and $p$-type doped (In)GaAsSb/GaSb as the source (Fig. 1a). This TFET structure is chosen to balance the on- and off-state performance, which has been systematically investigated in previous works including tuning source materials[17] and doping concentrations[19] as well as device scalability[30]. To achieve NTC behaviour, a gate/source overlap is employed by defining the gate metal (W) above the *nid*-InAs channel segment (Fig. 1b). The full device structure of a nanowire ferro-TFET with a gate-all-around architecture is illustrated in Fig. 1c. The details of the fabrication steps are described in Methods and Supplementary Fig. 1. The BTBT process dominating the carrier transport in the device has been confirmed by the negative differential resistance (NDR) obtained when reversing the source and drain bias of the device[12] (Supplementary Fig. 2a). Transconductance ($g_m$) in a FET is defined as $g_m = dI_D/dV_G$, and thus $g_m$ becomes negative when $I_D$ decreases with increasing $V_G$. Figure 1d shows the representative NTC ($g_m < 0$) with a peak-to-valley-current-ratio (PVCR, the $I_D$ ratio between the peak point (3) and the valley point (4)) of about 50 in the transfer characteristic of the ferro-TFET with the measurement setup shown in Fig. 1c. Statistics of NTC behaviour are shown in Supplementary Fig. 2b. Figure 1e elucidates the origin of the NTC in ferro-TFETs through the band diagrams related to different $V_G$ indicated in Fig. 1d. At point (1), the device is in the off-state where the BTBT is blocked due to the energy of the channel conduction band edge ($E_C$) being above the Fermi level of the source ($E_{F,s}$). When increasing $V_G$ up to point (2), the channel $E_C$ moves below $E_{F,s}$, leading to an accessible path for carriers to tunnel thereby increasing $I_D$. The BTBT transmission probability increases with the difference between channel $E_C$ and $E_{F,s}$ as continuously increasing $V_G$ until point (3) at which a strong source depletion starts near the heterojunction interface due to the gate/source overlap. This results in the source energy bands moving down which decreases the tunnelling transmission probability ($I_D$ saturation). Further increasing $V_G$ widens the source depletion region and closes the entire tunnelling path, causing the BTBT prohibition[21,31] and the valley $I_D$ at point (4).

The ferroelectricity in the HZO gate oxide enables reconfigurable NTC in the ferro-TFETs. Depending on the $V_{pulse}$ applied to the gate (+4 V/250 ns or −4 V/250 ns), the polarization in the ferroelectric film can be switched as shown in Fig. 1f, corresponding to the low-$V_{peak}$ or high-$V_{peak}$ state, respectively (inset of Fig. 1g; $V_{peak}$ is defined by $V_G$ at $I_D$ peak). Although the amplitude of $V_{pulse}$ can be substantially lowered by increasing the pulse width, ±4 V / 250 ns has been optimized to balance the required voltage amplitude and pulse width. Here, 13-nm-thick HZO is used as the gate oxide in our ferro-TFETs as such films exhibit robust ferroelectricity at a thermal budget of below 500 °C[32,33]. This is beneficial for III-V materials which may lack thermal stability at higher annealing temperatures. Transfer characteristics measured for the two distinct polarization states with evident ferroelectric hysteresis at various $V_{DS}$ are plotted in Fig. 1h, confirming the reconfigurability of our ferro-TFETs. A high quality of NTC with significant PVCR reaching a maximum value over 2 orders of magnitude (Supplementary Fig. 3) provides high symmetry of the transfer curves in the low-$V_{peak}$ state. In both states,

$V_{peak}$ positively shifts when $V_{DS}$ is increased (Fig. 1g) and a higher $V_G$ is needed to start suppressing the BTBT by moving down the bands in the gate-overlapped source region at a larger $V_{DS}$. The difference of $V_{peak}$ between the two states ($\Delta V_{peak}$) slightly increases with $V_{DS}$ and approaches 0.45 V at $V_{DS} = 0.5$ V (Fig. 1h). This value is somewhat small compared to other FeFET implementations but is mainly a result of memory window degradation after many switching cycles. Noticeably, the peak current is lower in the high-$V_{peak}$ state than that in the low-$V_{peak}$ state. The exact cause of this is not entirely understood but it may be caused by the different impact from the gate polarization on the source and channel region, which may further alter the factors that determine the maximum tunnelling current, such as the height and width of the tunnel barrier, and the density-of-states in the source region. Nevertheless, the difference in maximum $I_D$ will not change the conclusions of this work.

### Reconfigurable frequency doubling/phase shift

To realize frequency doubling, nonlinear devices such as transistors[34,35] and Schottky diodes[36,37] are typically employed. However, these conventional devices usually produce undesired harmonics that need to be suppressed by complicated circuits or additional filters, thus dramatically increasing the device area and power consumption in the system. One potential solution to overcome this challenge is to use single transistors with symmetric transfer characteristic for frequency doubling[9,11,38,39].

We here demonstrate reconfigurable frequency doubling by using the NTC in a single ferro-TFET. The basic operation is illustrated in Fig. 2b using the measurement setup scheme shown in Fig. 2a. First, the polarization states in the ferroelectric gate oxide can be programmed by $V_{pulse}$ (+4 V/250 ns or −4 V/250 ns). When an input sinusoidal signal ($v_{in}$) is applied to the gate and oscillates around a voltage near $V_{peak}$ (low-$V_{peak}$ state in Fig. 2b), each semi-cycle of $v_{in}$ (A–B–C or C–D–E) leads to a complete cycle (A–B–C) in the output current ($i_{out}$). As a result, the frequency of the output signal $i_{out}$ from the ferro-TFET will be doubled (low-$V_{peak}$ state in Fig. 2b) as compared to that of $v_{in}$. In the high-$V_{peak}$ state, however, with the same $v_{in}$, $i_{out}$ retains the input frequency as $v_{in}$ only operates in the $I_D$-$V_G$ branch with positive $g_m$ that shows nearly linear $I_D$-$V_G$ dependency, thus realizing frequency transmission.

The measured time-domain waveforms of $v_{in}$ and $i_{out}$ distinctly show frequency doubling and transmission reconfigured by +$V_{pulse}$ and −$V_{pulse}$, respectively (Fig. 2c). After pulsing +4 V/−4 V with 250 ns for 10 cycles, the $i_{out}$ waveforms have almost identical current levels with the expected output frequency ($f_{out}$), suggesting no performance degradation within 10 reconfiguration cycles. In communications and signal processing, digital data can be encoded as frequency shifting, where each frequency represents a digit, thus realizing the information encoding. For a binary frequency-shift keying (BFSK), the data is encoded as '1' and '0' in a square wave, mixing with a carrier wave with a certain frequency $f_{in}$ (Supplementary Fig. 4a), and the readout system translates the data to two discrete frequencies, for instance, $f_{out} = 2f_{in}$ for '1' and $f_{out} = f_{in}$ for '0' (Supplementary Fig. 4b). Here, other than typically using the square wave representing the dataset of '0's and '1's at the input, the data can also be written by $V_{pulse}$ (+4 V/250 ns for '1' and −4 /250 ns for '0', respectively), and the corresponding $f_{out}$ is then read as '1' (2 kHz) or '0' (1 kHz) as shown in Fig. 2c. In this case, the data is stored without application of external voltage due to the non-volatile ferroelectric polarization. Apart from the frequency doubling, the reconfigurable NTC behaviour can be also implemented as a non-volatile phase shifter. In this case, $v_{in}$ should oscillate within two peaks in the transfer curves (Fig. 2d). Due to the NTC in the right branch of the transfer curve in the low-$V_{peak}$ state, the output signal shifts its phase by 180° ($\Delta\varphi = 180°$) as compared to that of $v_{in}$ (Fig. 2e). In the high-$V_{peak}$ state, the output signal follows the identical phase ($\Delta\varphi = 0°$) as $v_{in}$ (Fig. 2e).

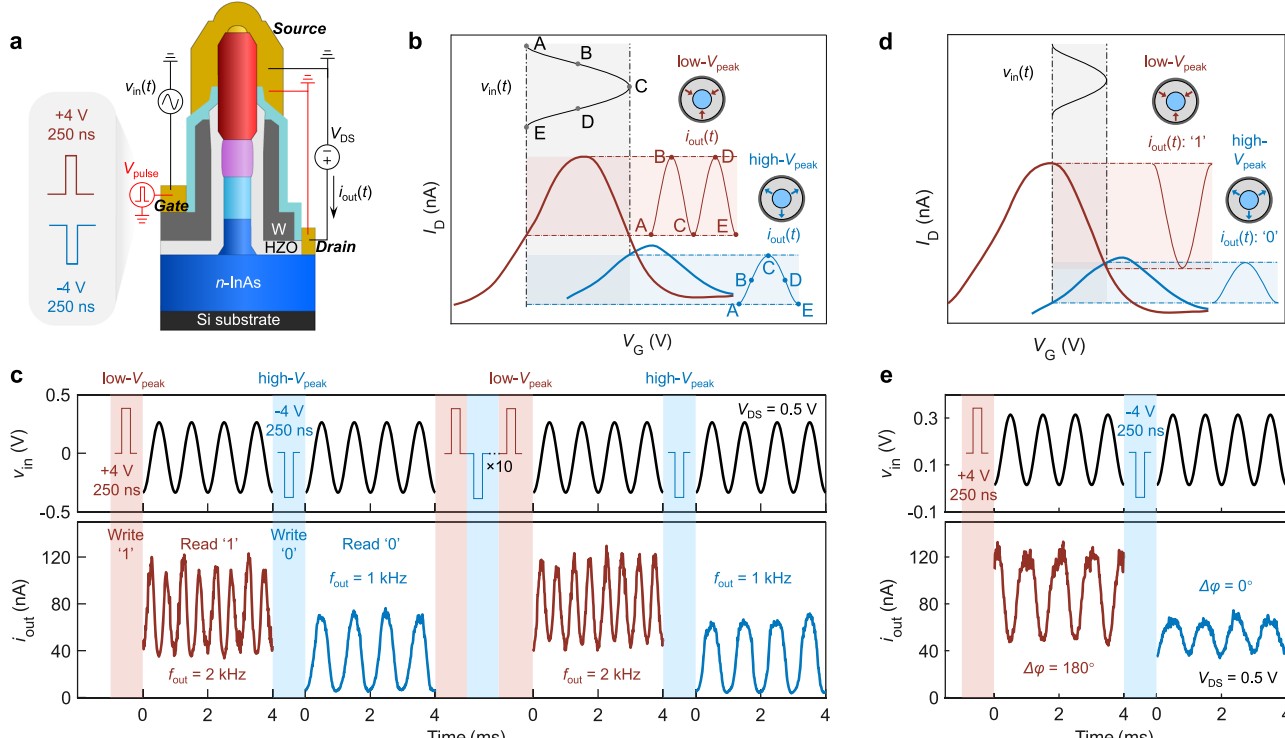

**Fig. 2 | Reconfigurable frequency doubling and phase shift in ferro-TFETs. a** The schematic of the electrical measurement setup with an alternating current (AC) signal as input for the ferro-TFET. The black scheme denotes the measured transfer characteristics while the red scheme indicates the gate voltage pulse ($V_{pulse}$) that sets the polarization in HZO gate oxide. Here, $+V_{pulse}$ (+4 V/250 ns) and $-V_{pulse}$ (−4 V/250 ns) represent binary '1' and '0', respectively. **b** The working principle for reconfigurable frequency doubling in the ferro-TFET. **c** Representative excerpt of the time-domain waveforms of $v_{in}$ (a sinusoidal wave with $f_{in}$ = 1 kHz) and $i_{out}$. The same result is obtained after 10-cycle reconfigurations. This can be used for BFSK to encode data as '1' and '0' in communication systems. **d** The working principle for reconfigurable phase shift in the ferro-TFET. **e** The demonstration of the excerpt of the time-domain $i_{out}$-$v_{in}$ for reconfigurable phase shift in ferro-TFETs.

Furthermore, the frequency doubling is accomplished with input frequency ($f_{in}$) up to 10 kHz with slight waveform distortion (Supplementary Fig. 5). The operational $f_{in}$ for the presented ferro-TFET device architecture is mainly limited by the large parasitic capacitance originating from the high-permittivity gate oxide between the electrode pads which are large compared to the nanowire channel region. For instance, the planar parasitic capacitance between the drain and gate electrode pad is about 5 orders of magnitude larger than the oxide capacitance at the nanowire channel, leading to dramatic suppression of operational frequency in the ferro-TFET. An optimized process with low-permittivity spacers such as hydrogen silsesquioxane (HSQ) or $SiO_2$ can mitigate the parasitic capacitances in vertical nanowire transistors, which can extend the operating frequency to GHz range[40,41] (detailed discussion in Supplementary Fig. 5). In fact, vertical III-V nanowire TFETs with similar device structure but with low-permittivity spacers has shown a cut-off frequency up to 3 GHz[42]. Despite the limit in high-frequency applications, low-frequency implementations (Hz - kHz) in IoT systems such as bio-signal sensing and modulation[43,44] can be practically realized by current ferro-TFETs. The benefit of low operational voltage in our ferro-TFETs enables low power consumption, in line with the requirement of IoT devices for such application schemes.

Due to the reconfigurability, the ferro-TFET can be preprogramed to either frequency doubling or frequency transmission in a communication system. In this application scheme, the retention time is important. Thus, we examine this in the ferro-TFET by inspecting the $I_D$ peak position, $V_{peak}$, in the two states as a function of time since $V_{peak}$ is critical to determine the waveform shape of $i_{out}$. The measurement was performed after stable device operation was obtained following the initial wake-up phase and $V_{peak}$ in the two reconfigurable states is retained for at least 20 days. The result of high-quality $i_{out}$ waveforms

shows that the frequency doubling still operates well 20 days after setting the state (Supplementary Fig. 6). Moreover, we also measure the endurance of NTC. The measured device shows an endurance of >10^5 pulsing cycles with stable $V_{peak}$ value in the low-$V_{peak}$ state (Supplementary Fig. 7), in line with other III-V ferroelectric integrations[32] and early Si implementations[45]. Although the measured endurance is on the low side thus making the proposed BFSK application challenging for our current ferro-TFET, it may be still useful in some special applications such as security systems in which disabling functions of the device is beneficial to complicate the reverse-engineering.

## Operation with parabolic transfer characteristic

Theoretically, ideal parabolic transfer characteristic leads to an ideal frequency doubler with 100% output power concentrated at $f_{out}$ = $2f_{in}$. Here, we analyse the power spectrum of reconfigurable frequency doubling in our ferro-TFET and evaluate the ideality of the $I_D$−$V_G$ parabolicity caused by the presence of NTC. We measure the output voltage ($v_{out}$) with an oscilloscope (Fig. 3a) by connecting the ferro-TFET to a resistor ($R$) in series and achieve nearly perfect waveforms of $v_{out}$ in both states with expected frequency when $f_{in}$ = 1 kHz (Fig. 3b). Notably, the amplitude of $v_{out}$ in the high-$V_{peak}$ state is larger due to a wider operating current range in the $I_D$−$V_G$ curve compared to that in the low-$V_{peak}$ state when applying an input signal near the current peak in the low-$V_{peak}$ state, which might lead to slightly different cut-off frequencies in two states. However, this unevenness will not affect the function of frequency modulation since the frequency component of $v_{out}$ in two states is more critical in this application scheme. The power spectrum of the output signal shows that a centre frequency of 2 kHz has ~98% of the output power concentration in the low-$V_{peak}$ state (Fig. 3c). In the case of the high-$V_{peak}$ state, almost all (99.6%) of the output power is centred at 1 kHz (Fig. 3d). The power spectrum in dBm

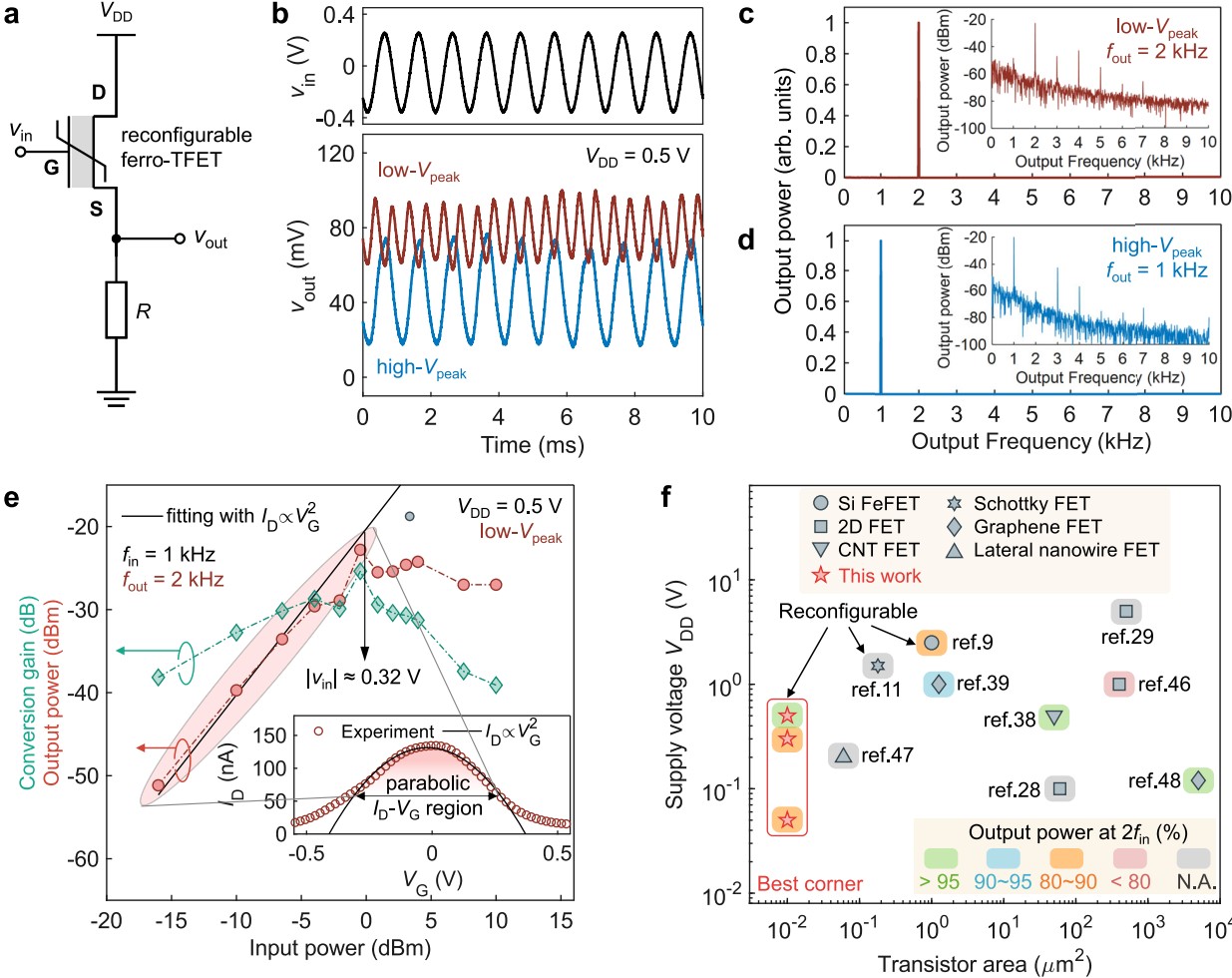

**Fig. 3 | Spectral analysis and low-power operation of reconfigurable ferro-TFETs. a** The schematic of measurement setup for output voltage ($v_{out}$) waveform. D, G and S denote drain, gate, and source of the ferro-TFET, respectively. The resistor $R = 8\,M\Omega$. **b** Representative excerpt of the time-domain waveforms of $v_{in}$ and $v_{out}$ in two reconfigurable states. **c, d** The output power spectra of $v_{out}$ in the low-$V_{peak}$ (**c**) and high-$V_{peak}$ state (**d**), respectively. The inset shows the corresponding power spectrum in dBm. All the frequency spectra are obtained by applying the fast Fourier transform (FFT) algorithm with filtering the DC components. **e** The output power (at $f_{out} = 2\,kHz$) and the conversion gain of the ferro-TFET as a function of input power (at $f_{in} = 1\,kHz$) in the frequency doubling mode. The output power shows great overlay with the ideal parabolic $I_D$–$V_G$ fitting (black line) with respect to the input power up to ~0 dBm (amplitude of $v_{in}$: $|v_{in}| \approx 0.32\,V$). The inset shows the corresponding $I_D$–$V_G$ curve fitting with the ideal parabola ($I_D \propto V_G^2$). **f** Benchmarking of this work against other single-transistor frequency doublers. The transistor area from literature is based on the product of channel width and length or the estimation from the top-view microscope image. The operating frequency reported in listed work ranges from 10 Hz to 200 kHz. N.A. denotes "not available".

shows that unwanted harmonics have >20 dB lower than the desired frequency in both frequency doubling and transmission mode (insets of Fig. 3c, d), indicating that our ferro-TFET-based frequency doubler possesses a very high spectral purity with significant suppression of undesirable harmonics without the use of additional filters in both reconfigurations.

We attribute this high spectral purity for frequency doubling to the shape of transfer curve in the ferro-TFET. By tuning the amplitude of $v_{in}$, the output power and the conversion gain are obtained as a function of input power in Fig. 3e. The output power approaches the saturation at ~0 dBm of the input power where the conversion gain accordingly reaches the maximum. The result of output power fits well with the ideal parabolic $I_D$–$V_G$ relationship for over 15-dB dynamic range. The inset of Fig. 3e indicates that an input signal with an amplitude up to ~0.35 V (~0.8 dBm in power) can operate with a perfect parabolic $I_D$–$V_G$, in agreement with the saturation point (~0.32 V) shown in Fig. 3e. Thus, an amplitude of 0.3 V used for $v_{in}$ in this work allows the input signal operating in the range with the ideal case of a concave parabola as well as a nearly maximum conversion gain. These

features verify and explain the fact that ferro-TFETs as frequency doublers produce highly pure output frequencies, particularly for weak signal processing in low-power systems, which has much smaller voltages than those used in ambipolar FETs[9,11]. This differs from ambipolar TFETs[22,23] which have exponential $I_D$–$V_G$ dependence around the current valley in the subthreshold region. Due to this, NTC-based ferro-TFETs may double the frequency of a small input signal with a higher spectral purity at the output compared to ambipolar TFETs.

To obtain high conversion gain at desired frequency, power- and area-hungry filtering circuits are usually employed in conventional frequency doublers. In order to shrink area, there are many single-transistor frequency doublers[9,11,28,29,38,39,46–48], but very few can operate with a $V_{DD} < 1\,V$ while retaining a scaled footprint per device below $1\,\mu m^2$, thereby being challenging for low-power systems and high-density integration with other technologies on the same platform. Area scaling also lowers the power consumption as it reduces the intrinsic capacitive contribution of the device. As a result of the parabolic transfer curve due to the robust NTC properties (Supplementary Fig. 3), spectral purity remains high when further reducing the drive

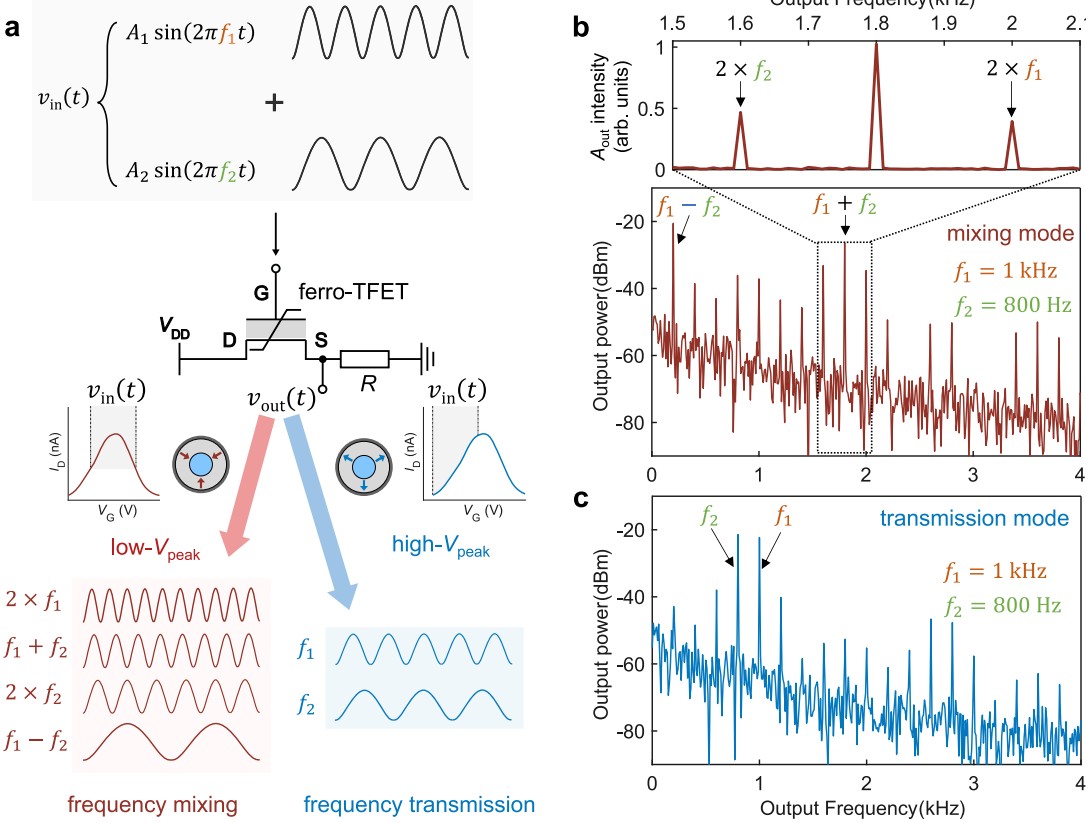

**Fig. 4 | Reconfigurable frequency mixing in a single ferro-TFET. a** Working principle of the reconfigurable frequency mixing controlled by the polarization in the ferroelectric gate oxide. **b**, **c** The power spectrum in (**b**) mixing mode and (**c**) transmission mode in the low-$V_{peak}$ state and high-$V_{peak}$ state, respectively. The magnified figure of (**b**) shows that the $v_{out}$ amplitude ($A_{out}$) intensity at $f_1 + f_2$ is almost twice that at $2f_1$ or $2f_2$, in agreement with the calculation result in Supplementary Note 1. Here, $A_1 = A_2 = 0.3$ V, $f_1 = 1$ kHz, $f_2 = 800$ Hz, $V_{DD} = 0.5$ V, and $R = 8$ MΩ.

voltage, reaching 88% at $V_{DD} = 0.3$ V and 83% at ultra-low $V_{DD} = 50$ mV (Supplementary Fig. 8). The slight reduction in spectral purity at lower $V_{DD}$ is due to the nonlinear change in the $I_D$-$V_G$ characteristic with $V_{DD}$ (detailed discussion in Supplementary Fig. 8). Combined with the aggressively reduced footprint using the vertical nanowire architecture, our ferro-TFET stands out from other reported single-transistor frequency doublers towards the best corner of energy- and area-efficiency while possessing high power concentration at target $f_{out}$ and reconfigurability (Fig. 3f).

### Reconfigurable frequency mixing

A frequency mixer is a nonlinear device that produces new signals corresponding to the sum and difference of the original frequencies from the input signal, and is widely used for transmitters (frequency up-conversion) and receivers (frequency down-conversion) in wireless communications[49]. We propose and demonstrate reconfigurable frequency mixing in a single ferro-TFET. Figure 4a illustrates the operation principle of the most common case in which the input signal consists of two different frequencies ($f_1$ and $f_2$). Given the reconfigurability of the ferro-TFET, two different functionalities can be realized. In the low-$V_{peak}$ state, multiple output frequencies are generated in which the ferro-TFET acts as a conventional frequency mixer provided by the parabola-shaped transfer curve. For an ideal parabolic transfer curve, $v_{out}$ is proportional to $v_{in}^2$, leading to four dominant frequencies created: $f_1 - f_2$, $2f_1$, $f_1 + f_2$, and $2f_2$ (derivation in Supplementary Note 1). In contrast, in the high-$V_{peak}$ state, the input frequencies are transmitted to the output as the device only operates in the positive $g_m$ branch of the transfer curve.

We experimentally implement frequency mixing on the ferro-TFET by applying $v_{in}$ summed by two sinusoidal signals with $f_1 = 1$ kHz and $f_2 = 800$ Hz, respectively, and for simplicity with the same amplitude ($A_1 = A_2$). The collected time-domain $v_{out}$ waveforms in two reconfigurable modes (Supplementary Fig. 9) are complex and their frequencies are not straightforwardly recognizable. Instead, the corresponding power spectrum in mixing (low-$V_{peak}$ state) and transmission mode (high-$V_{peak}$ state) provides rich information regarding the output signal as shown in Fig. 4b, c, respectively. In the mixing mode, most of the output power (~90%) is concentrated at $f_1 - f_2$ and $f_1 + f_2$, suggesting an excellent spectral purity with negligible undesired harmonics (Fig. 4b), thus being applicable for both down- and up-conversion. In the transmission mode (high-$V_{peak}$ state), the spectrum is dominated by two input frequencies, $f_1$ and $f_2$, while other frequencies such as $2f_2 \pm f_1$ and $2f_1 \pm f_2$ are considerably suppressed (Fig. 4c), validating the realization of frequency transmission for a two-frequency signal. Notably, the $v_{out}$ amplitude intensity of $f_1 + f_2$ is almost double that of $2f_1$ and $2f_2$ (top of Fig. 4b), in good agreement with the ideal parabolic case where $v_{out}$ is proportional to $v_{in}^2$ (Supplementary Note 1). This again confirms nearly perfect parabolic $I_D$-$V_G$ around the current peak (Fig. 3e) existing in our ferro-TFETs owing to the robust NTC.

## Discussion

Many advantages of using ferro-TFETs for signal modulation are identified. The first significant benefit of the ferro-TFET over conventional modulation device is its reconfigurability enabled by the ferroelectric polarization in the gate oxide, which brings various

functionalities in the same device. This may find interest in hardware security[4,6] of analogue chips where the true function of the device can be well hidden in the design. Secondly, an ultra-low $V_{DD}$ down to 50 mV can be used to drive the device, thereby providing ferro-TFETs with potentials in energy-efficient wireless communications. Although the operating frequency is not yet optimized for high-frequency applications, many IoT systems with low frequencies such as bio-signal sensor[44] can be implemented. Compared to conventional transistor-based doublers/mixers, the ferro-TFET doubler/mixer generates negligible harmonics without filters owing to the parabolic $I_D$-$V_G$ curve obtained by the robust NTC. Furthermore, a single vertical nanowire architecture reduces the footprint of the device compared to its planar counterparts (Fig. 3f), being advantageous for high-density integration. For instance, an optimized circuit usually used for mixers to suppress harmonics consisting of six planar FETs[50] has an area above 500 μm² while our ferro-TFET device cell only requires ~0.01 μm². This will considerably simplify the potential analogue circuit design while maintaining the conversion gain. All these benefits point towards a lower-power, higher-density, and more functional communication system.

In summary, we demonstrate a number of reconfigurable operations based on signal modulation implemented by a single nanowire ferro-TFET. In our structure, a gate/source overlap creates a strong NTC in the ferro-TFET, leading to a highly parabolic transfer characteristic, which can be used for frequency doubling without generating additional harmonics. By switching the ferroelectric polarization in the HZO gate oxide, the transfer curve significantly shifts while retaining the parabolicity, creating two distinct states (low- and high-$V_{peak}$ state). Depending on the bias of input signal, both reconfigurable frequency doubling/transmission (Fig. 2b) and phase shift (Fig. 2d) are demonstrated. Our ferro-TFET also shows high reliability for both polarization states including high endurance and long measured retention time with stable $V_{peak}$. More sophisticated frequency modulation such as two-analogue-signal frequency mixing has also been implemented in the single ferro-TFET with excellent suppression of undesired harmonics. Our results indicate that various signal modulation schemes can be realized and reconfigured in a nanoscale device unit with ultra-low operational voltage, significantly increasing the functional diversity and energy efficiency. Furthermore, in line with the goal of hyper-scaling for the future electronics[1], the presented fabrication flow of ferro-TFET adapts well to that of the state-of-the-art TFET logic devices using similar III-V structures[15], and thus, a hybrid low-power platform including steep-slope logic devices without NTC and reconfigurable frequency modulation devices with NTC can be achieved on a single semiconductor die with the same processing scheme[30]. This monolithic integration decreases the complexity and feature sizes for mixed signal circuit design and digital/analogue coupled modules utilized in emerging technologies such as IoT[8] and quantum computing[51].

## Methods

### Device fabrication

The sample was initialized by the nanowire growth by metal-organic vapour-phase epitaxy (MOVPE) via Au-assisted vapour-liquid-solid (VLS) process. Prior to the growth, Au dots were prepatterned by electron-beam lithography on a 260-nm highly-doped InAs buffer layer on the Si substrate. Next, highly $n$-type doped InAs, $nid$-InAs as well as $p$-type (In)GaAsSb and $p$-type GaSb segments were sequentially grown for the drain, the channel, and the source, respectively, as illustrated in Supplementary Fig. 1a. Sn and Zn were utilized as the $n$- and $p$-type dopant for the drain and source, respectively.

Supplementary Fig. 1 summarizes the processing flow of the ferro-TFET. The device fabrication started with the digital etch (DE) which reduces the channel diameter[52] and removes the oxide states at the channel interface thus improving the electrostatics[53,54]. Here, 3

cycles of DE were performed by sequentially repeating the ozone exposure and wet etch in citric acid. Next, 13-nm HZO was grown at 200 °C for the gate oxide by thermal atomic layer deposition with a 1:1 alternation between the precursors TDMAHf and TEMAZr using water as the oxygen source. The thickness of the film was confirmed by ellipsometry and structure details such as polycrystalline characteristic and interface quality of HZO on InAs have been verified[32]. A 60-nm-thick W gate metal was then sputtered and aligned above the $nid$-InAs/$p$-(In)GaAsSb heterostructure using a UV-lithography S1813 (photoresist) mask and back-etch in a reactive ion etch (RIE) system. The exposed W on the top of the nanowire was etched by plasma $SF_6$:Ar in the same RIE chamber (Supplementary Fig. 1b) to define the gate length, leading to the gate/source overlapping (Fig. 1b). The top HZO was wet etched by HF 1:400 to expose the source contact region (Supplementary Fig. 1c) using the S1813 mask. The sample was annealed at 450 °C for 30 s in a $N_2$ ambient using rapid thermal annealing (RTA) for crystalizing the HZO into the ferroelectric orthorhombic phase. The sample was finally metallized for contacts (Ni/Au) and 10-nm $Al_2O_3$ was used as the top spacer to isolate the gate and source (Supplementary Fig. 1d).

### Electrical characterizations

The electrical characterizations were performed in a MPI TS2000-SE probe station using a Keysight B1500A Parameter Analyzer. A B1530A waveform generator module (WGFMU) was utilized for both the pulsed $I$-$V$ measurements ($I_D$-$V_G$ transfer characteristics) and fast voltage pulses ($V_{pulse}$) to switch the ferroelectric polarization in the HZO gate. Additionally, the time-domain sinusoidal waves ($v_{in}$) and output drain current ($i_{out}$) are generated and sensed by B1530 WGFMU modules, respectively. The waveforms of $v_{out}$ were collected by the Rohde and Schwarz RTO Digital Oscilloscope with a conventional common-drain circuit configuration as shown in Fig. 3a. For the signal mixing measurement presented in Fig. 4, a Rigol DG1000Z function generator was used to create two sinusoidal signals with different frequencies from two channels which were summed by using a BNC T-adapter (2-in 1-out). The output signal from the T-adapter is $v_{in}$ used in the signal mixing measurements. The crosstalk between two input signals is negligible (detailed discussion in Supplementary Fig. 9). All power spectra of $v_{out}$ were obtained by applying the fast Fourier transform (FFT) algorithm on the collected waveforms directly from the oscilloscope with filtering of the DC component. To achieve high enough resolution of frequency-domain power spectra, 100 periods of all waveforms were captured from the oscilloscope.

## Data availability

The data that support the findings of this work are available within the article and its Supplementary Information file. Source data are provided with this paper.

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

## Acknowledgements
This work was supported in part by the project DYNAMISM (Dynamic Properties of Ferroelectric III–V MOSFETs) under the European Research Council (ERC) Advanced grant with a reference number of 101019147 (L.-E.W.) and in part by the Swedish Research Council (VR) under Grant 2016-06186 (Electronics beyond kT/q, L.-E.W.). The authors would like to thank Dr. Johannes Svensson and Gautham Rangasamy from Lund University for the help of electrical characterization with the oscilloscope.

## Author contributions
Z.Z., A.E.O.P., and L.-E. W. conceptualized the work. Z.Z. performed the sample growth and fabrication. Z.Z. and A.E.O.P. carried out the electrical characterizations. Z.Z. analysed and visualized the data. Z.Z. wrote the original manuscript. L.-E.W. supervised the work. All the authors discussed and revised the final manuscript.

## Funding

## Competing interests
The authors declare no competing interests.
