## [Peer Review File · Nature Communications]

Reconfigurable Signal Modulation in a Ferroelectric Tunnel Field-Effect TransistorREVIEWER COMMENTS

Reviewer #1 (Remarks to the Author):

In this manuscript, the authors report the design and demonstration of a reconfigurable analog signal modulation based on ferroelectric tunnel Field-effect transistors (FETs). A gate/source overlapping channel with a heterostructure was designed, and the authors demonstrated parabolic transmission characteristics with reconfigurable performance by introducing ferroelectric materials as the gate dielectric. The authors also showed that the same analog signal modulation function as reported in previous works can be achieved using the same circuit structure. [1-6] However, the significance on reducing the operating voltage is not clearly explained, compared to previous works, and the working speed of the circuit (KHz) is below the requirement for most practical applications, which is typically above MHz as demonstrated in previous works. [1-6] Therefore, I cannot recommend this manuscript to be published in Nature Communications. Besides, I have the following concerns that are required to be addressed.

1. The reliability of the nonvolatile regulation is important for reconfigurable electronics. But I cannot find the relevant data in the paper. The authors need to provide data on the retention and endurance of the device to prove that the device has excellent state retention characteristics and stable working ability, after a large number of erase and write operations.
2. The proposed mechanism is not solid to me. It is necessary to provide more experimental evidences to support the proposed proposing of the BTBT mechanism. For instance, the measurement of temperature-dependence of the field-effect transfer characteristics should be implemented.
3. The working speed of the circuit is a critical parameter for practical applications, while the demonstrated working speed is too slow to be useful in practical applications. In principle, the enhancement of the peak current of the device would be helpful. Probably, the authors can consider optimizing the device design to increase the peak current (currently only 100 nA).
4. For the application of the analog signal modulation, it is expected to simultaneously achieve low voltage operation and less harmonic power. However, it is not the case, as shown in Supplementary Figure S7, where the increase in harmonic power with the decrease of V_{dd} . The authors should explain the reason more clearly. Whether the harmonic power can be kept unchanged with the reduction of V_{dd} ?
5. The circuit architecture used in the manuscript seems completely the same as the previous works. The authors are suggested to add more data or discussion to highlight the difference.
6. The detailed measurement processes are missing. For example, the authors did not provide the method for realizing signal mixing in Figure 4. May the possible crosstalk between two input signals result in degradation of test results? If not, authors need to explain this. The method of power spectrum data was not described. It is not clear whether the data was obtained directly from oscilloscope or software analysis. If software analysis was used, the number of test cycles should be specified.
7. The hysteresis characteristic in the measured field-effect transfer curve is one of the important criteria to evaluate the quality of ferroelectric gate FET devices, which was displayed in the manuscript.
8. What is the theoretical limit of the proposed device, which is one of my concern in considering its promising application in signal modulation. Authors are suggested to further reduce the voltage value and pulse width of the voltage pulse to operate the device state, and explore the limit.
9. The yield and uniformity of the electrical characteristic are critical to evaluate the promise of the novel device, and therefore authors should provide the related experimental data.

References:

- [1] Science 332, 1294 (2011)
- [2] Nano Lett. 15, 1883 (2015)
- [3] IEEE Electron Device Lett. 30, 547 (2009)
- [4] Sci. Rep. 7, 46605 (2017)
- [5] Nat. Commun. 3, 1018 (2012)
- [6] ACS Nano 4, 10, 5532 (2010)

Reviewer #2 (Remarks to the Author):

The paper proposes to use ferroelectric TFET for reconfigurable analog applications. Leveraging the parabolic ID-VG characteristics of a ferro-TFET with gate/source overlapped channel, the authors demonstrated its application in frequency-doubling, 180 degree phase shift, signal transmission, and frequency mixing. The work is interesting and shows applications of ferroelectric devices in reconfigurable analog applications. The reviewer has a few recommendations.

1. The authors used the negative transconductance to achieve parabolic ID-VG characteristics. It is known that TFT shows ambipolar transport, which also features two branches. The authors are encouraged to further clarify the pros and cons of their design. The relevant reference of Fe-TFT are:

- [1] J. Luo et al., "A Novel Ambipolar Ferroelectric Tunnel FinFET based Content Addressable Memory with Ultra-low Hardware Cost and High Energy Efficiency for Machine Learning," VLSI Symp. 2022
- [2] J. Luo et al., "Novel Ferroelectric Tunnel FinFET based Encryption-embedded Computing-in-Memory for Secure AI with High Area-and Energy-Efficiency," IEDM 2022

2. The authors are encouraged to show the switching speed of the device. In this work, only +/-4V, 250ns write pulse is used. What is the switching speed of this device? With 13nm HZO, the hysteresis window of less than 0.5V is ultra-small. Could the authors comment on this?

3. On page 6, the authors mentioned the application of the device in frequency shift keying. Along with many applications in Fig.2 and Fig.3, the modulated output amplitudes are also changing. This is because the high-Vpeak and low-Vpeak has a different amplitudes. How does the unequal amplitude impact the overall performance of the applications? In addition, the authors mentioned to program the device as a way of encoding the digital data. This brings a significant challenge to device endurance. The authors need to clarify that.

4. Fig.3b shows a smaller output amplitudes for low-Vpeak than the high-Vpeak, which seems contradictory to Fig.1g.

5. The Fig.5f benchmarking is not entirely fair. It only reflects a partial picture. How about the operation frequency? Though the authors claim that similar device shows a cutoff frequency of 3GHz on page 7, it does not mean that the current device will has that performance given that the oxide capacitance is much thicker for ferroelectric. Even though the same cutoff frequency carries over for ferroelectric device, the speed of the device is still much smaller than its FeFET counterpart. For analog application benchmarking, the area may not be the only metric of interest.

6. In the discussion, the authors claim that the VDD can be as low as 50mV, which could be confusing. It just shows that the drain supply voltage can be low. The gate voltage is still much larger. In addition, the necessity of +/-4V programming also brings significant

overhead.

Reviewer #3 (Remarks to the Author):

In this paper, author designed a reconfigurable ferroelectric TFET. And the diverse functionalities of signal modulations including frequency transmission, doubling, mixing, and phase can be achieved in this single device. The context and topic in this paper is interesting, but the author needs to further revise the article for the following problems:

1. The author only analyzed the working principle and test results of the device, and needs to further analyze the influence of structure parameters on device performance.
2. The author should give detailed parameters of ferroelectric material HZO in the paper.
3. The author mentioned in the paper that the peak current is lower in the high- V_{peak} state than that in the low- V_{peak} state. The exact cause of this is also analyzed.

Response Letter to Reviewers

We thank all reviewers for providing useful comments and suggestions to help improve this work. We have now replied to all questions and comments point by point in the following, and substantially revised the manuscript accordingly. All changes in both the main manuscript and Supplementary Information can be found highlighted in yellow.

To Reviewer #1:

In this manuscript, the authors report the design and demonstration of a reconfigurable analog signal modulation based on ferroelectric tunnel Field-effect transistors (FETs). A gate/source overlapping channel with a heterostructure was designed, and the authors demonstrated parabolic transmission characteristics with reconfigurable performance by introducing ferroelectric materials as the gate dielectric. The authors also showed that the same analog signal modulation function as reported in previous works can be achieved using the same circuit structure. [1-6] However, the significance on reducing the operating voltage is not clearly explained, compared to previous works, and the working speed of the circuit (KHz) is below the requirement for most practical applications, which is typically above MHz as demonstrated in previous works. [1-6] Therefore, I cannot recommend this manuscript to be published in Nature Communications. Besides, I have the following concerns that are required to be addressed.

We thank for the reviewer's summary and comments. The argument for reducing operating voltage follows the increasing importance for power reduction. Innovations that reduce power consumption and/or enhance integration density (both of which is achieved here) are essential and TFETs are considered a main contender for overcoming the Boltzmann tyranny and achieving low-power electronics [1]. In addition to the voltage scaling, device scaling is of equal importance as it will lower the required currents and the intrinsic capacitance.

We also acknowledge that the presented devices do not operate at MHz frequencies, however, we do not consider this a major issue as they are proof-of-concepts, and the cause of sub-MHz frequencies is well understood and can be solved by known solutions. As we discussed in the manuscript, the main factor limiting the speed is the excessive parasitic capacitance [2] existing between the electrodes as this process has not been optimized for frequency performance with a low- k bottom spacer (between gate and drain) such as HSQ or SiO₂. This well-established processing technique [3-5] will reduce the capacitance and is expected to increase the operating frequency well into the MHz-range. Notably TFETs are demonstrated to operate into the GHz frequency regime [5] with projections up to high mm-wave frequencies (>100 GHz) [2].

Furthermore, there are plenty of low-frequency applications in, for instance, bio-signal sensing (which is typically in the range of Hz ~ kHz) and low-data-rate IoT. Due to the low-power operating feature of TFETs, we therefore mainly consider our ferro-TFETs for low-power IoT applications. Thus, although the current version of our devices may not be suitable for most high-performance communication applications, it is perhaps already qualified for IoT applications with low operating frequencies and supply power. Cointegrating these low data rate reconfigurable devices with low-power TFET logic (very similar processing) would enable new capabilities for low-power applications.

Compared to the previous work refs. [1-6] that the reviewer listed, we think that our device has advantages in 1) low operating voltage ≤ 0.5 V (energy efficiency); 2) small device area (better high-density integration); 3) non-volatile reconfigurability (multiple functions). The operating frequency is lower than the listed references, but high frequency is not a requirement for some IoT applications.

We now start to address our concerns point-by-point for the following questions/comments.

References:

- [1] Datta S., *et al.* Toward attojoule switching energy in logic transistors. *Science*, 2022, 378(6621): 733-740.
- [2] Asbeck P. M., *et al.* Projected performance of heterostructure tunneling FETs in low power microwave and mm-wave applications. *IEEE Journal of the Electron Devices Society*, 2015, 3(3): 122-134.
- [3] Johansson S., *et al.* High-Frequency Gate-All-Around Vertical InAs Nanowire MOSFETs on Si Substrates. *IEEE Electron Device Letters* 35, 518-520 (2014).
- [4] Kilpi O. P., *et al.* Vertical nanowire III–V MOSFETs with improved high-frequency gain. *Electronics Letters* 56, 669-671 (2020).
- [5] Hellenbrand M., *et al.* Capacitance Measurements in Vertical III–V Nanowire TFETs. *IEEE Electron Device Letters* 39, 943-946 (2018).

1. The reliability of the nonvolatile regulation is important for reconfigurable electronics. But I cannot find the relevant data in the paper. The authors need to provide data on the retention and endurance of the device to prove that the device has excellent state retention characteristics and stable working ability, after a large number of erase and write operations.

We thank the reviewer for pointing this out and we agree that the reliability of the non-volatile functionality is important. In fact, we have investigated this property. The excellent retention of the presented device is highlighted in original Fig. S5, and is measured to be at least 20 days (10^6 s). The retention characteristic is measured after the initial wake-up phase when the cycle-to-cycle variation has stabilized, i.e., after a large number of erase and program operations compared to the endurance of our devices. The endurance, highlighted in original Fig. S6, is measured on another device on the same sample and reaches $>10^5$ which is a state-of-the-art value compared to other III-V integrations and similar to early Si adaptations. III-V ferroelectric integration is a nascent technology and has not yet developed to the levels achieved by Si, but with defect and interface engineering there is no reason to believe that it will not be possible to achieve similar performance.

To clarify this, we added this corresponding information about device reliability in lines 175-184 on page 8 of the revised manuscript highlighted in yellow as well as below:

In this application scheme, the retention time is important. Thus, we examine this in the ferro-TFET by inspecting the I_D peak position, V_{peak} , in the two states as a function of time since V_{peak} is critical to determine the waveform shape of i_{out} . The measurement was performed after stable device operation was obtained following the initial wake-up phase and V_{peak} in the two reconfigurable states is still retained for at least 20 days. ~~and~~ The result of high-quality i_{out} waveforms shows that the frequency doubling still operates well 20 days after setting the state (Supplementary Fig. 6). Moreover, we also measure the endurance of NTC. The measured device shows an ~~good~~ endurance of $>10^5$ pulsing cycles with stable V_{peak} value in the low- V_{peak} state (Supplementary Fig. 7), in line with other III-V ferroelectric integrations³² and early Si implementations⁴⁵.

2. The proposed mechanism is not solid to me. It is necessary to provide more experimental evidences to support the proposed proposing of the BTBT mechanism. For instance, the measurement of temperature-dependence of the field-effect transfer characteristics should be implemented.

We thank the reviewer for this question, and we admit that the evidence to prove the BTBT mechanism of the presented device is partly missing. A typical feature of a tunnel field-effect transistor (TFET) is that there is a negative differential resistance (NDR) effect when the source and drain is biased in the reverse direction [1]. This is attributed to a tunnel diode in the transistor, which verifies the BTBT mechanism for the device. We actually carried out this type of measurement on the presented device but didn't show the result in the original manuscript as this is not highly relevant to the main message of the work. This TFET structure has a well-established working mechanism presented in our previous works which we cited in the original manuscript, see refs. 15-17 in the original manuscript.

Regarding the temperature-dependence measurements, a previous TFET work of ours using an identical nanowire heterostructure has already been published [2], and we therefore find that this type of measurement is not required to prove the BTBT mechanism as the NDR effect has been observed.

Nevertheless, we realize that further information is needed for the readers. Therefore, we now provide the NDR result of the same device that we used in this work in the revised Supplementary Information Fig. 2a as well as below:

Supplementary Fig. 2. Negative differential resistance (NDR) in ferro-TFETs and device reliability.
a, The corresponding negative differential resistance (NDR) from the same ferro-TFET used in this work for reconfigurable signal modulation. The NDR behaviour is obtained by reversing the source and drain bias in the measurement. The result of NDR indicates a high-quality tunnel junction within the device and confirms that band-to-band tunnelling dominates the carrier transport.

We have also added the corresponding description including NDR measurements in lines 86-88 on page 4 of the revised manuscript highlighted in yellow as well as below:

The BTBT process dominating the carrier transport in the device has been confirmed by the negative differential resistance (NDR) obtained when reversing the source and drain bias of the device¹² (Supplementary Fig. 2a).

3. The working speed of the circuit is a critical parameter for practical applications, while the demonstrated working speed is too slow to be useful in practical applications. In principle, the enhancement of the peak current of the device would be helpful. Probably, the authors can consider optimizing the device design to increase the peak current (currently only 100 nA).

We fully agree with the reviewer that speed is important, but as stated above we do not find it crucial in this

proof-of-concept manuscript. As the reviewer points out, one of the possible solutions to increasing the speed is to increase the peak current, which could be achieved through modifying the composition of the source material [3] or the doping in the source [4]. However, the increase of peak current typically leads to degradation of off-state performance such as higher subthreshold swing and off-current [3,4]. Here, we chose a trade-off that would enable cointegration of low-power logic (i.e., steep-slope transistors) and analogue applications using the same nanowire heterostructure on the same sample (which would be greatly simplified if on- and off-state performance is balanced). The current application scheme only uses the on-state of the device for signal modulations, so the trade-off is in principle not required and a higher peak current is thus possible.

However, as mentioned above, the operating frequency of the current device is mainly limited by the huge parasitic capacitances existing in the device, especially by the parasitic capacitance between the planar HZO film and InAs buffer at the drain side (bottom), as we mentioned in lines 144-149 on page 7 of the original manuscript:

The operational f_{in} for the presented ferro-TFET device architecture is mainly limited by the large parasitic capacitance originating from the high-permittivity gate oxide between the electrode pads which are large compared to the nanowire channel region. For instance, the planar parasitic capacitance between the drain and gate electrode pad is about 5 orders of magnitude larger than the oxide capacitance at the nanowire channel, leading to dramatic suppression of operational frequency in the ferro-TFET.

Therefore, a MHz operating frequency can be achieved by inserting spacers with low permittivity such as HSQ or SiO₂ to reduce this parasitic capacitance. As we estimated in the original Supplementary Information (original caption of Fig. S4), the maximum operating frequency can be increased up to GHz if the parasitic capacitance becomes negligible. In fact, our previous work with a similar TFET structure but with a low-permittivity bottom spacer demonstrated a cut-off frequency up to 3 GHz (as we mentioned in the original manuscript in lines 151-152 on page 7). Here, as a proof-of-the-concept, we chose a simple device structure to demonstrate the reconfigurable signal modulation functionality rather than optimizing the high-frequency operation.

To clarify, we have added more details about how to improve the operational frequency in Supplementary Fig. 5 caption of the revised Supplementary Information highlighted in yellow as well as below:

The estimated parasitic capacitance between gate and the drain is $\sim 10^5$ larger than that of the oxide capacitance. Therefore, ~~By considering this~~ by inserting low-permittivity spacers such as hydrogen silsesquioxane (HSQ) or SiO₂ between electrodes, the limited operational frequency ~~can~~ may be extended up to 1 GHz. Changes in the nanowire heterostructure, such as modifying the composition of the source material or the doping concentration, may further increase the peak current to the benefit of higher frequency operation by negating the parasitic effect.

We have also modified accordingly in lines 165-168 on page 8 of the revised manuscript highlighted in yellow as well as below:

An optimized process with low-permittivity spacers such as hydrogen silsesquioxane (HSQ) or SiO₂ can mitigate the parasitic capacitances in vertical nanowire transistors, which can extend the operating frequency to GHz range^{40,41} (detailed discussion in Supplementary Fig. 5).

As mentioned by the reviewer, the working speed of the current ferro-TFET is slow and may be unsuitable for most of the communication applications. However, with the great benefit of low operating voltage in our device, we think that the ferro-TFETs may still find low-frequency-based IoT applications such as acquisition and modulation of bio-signals [5,6] which typically show low operating frequencies (Hz ~ kHz), as we

discussed in the beginning as well as the original manuscript in lines 230-232 on page 11:

Although the operating frequency is not yet optimized for high-frequency applications, many IoT systems with low frequencies such as bio-signal sensor⁴² can be implemented.

In such a bio application scheme, our ferro-TFETs would be suitable thanks to their low power characteristic. Therefore, we added corresponding discussion about the potential implementations with low-frequency systems also in lines 169-173 on page 8 of the revised manuscript highlighted in yellow as well as below:

Despite the limit in high-frequency applications, low-frequency implementations (Hz ~ kHz) in IoT systems such as bio-signal sensing and modulation^{43,44} can be practically realized by current ferro-TFETs. The benefit of low operational voltage in our ferro-TFETs enables low power consumption, in line with the requirement of IoT devices for such application schemes.

4. For the application of the analog signal modulation, it is expected to simultaneously achieve low voltage operation and less harmonic power. However, it is not the case, as shown in Supplementary Figure S7, where the increase in harmonic power with the decrease of V_{DD}. The authors should explain the reason more clearly. Whether the harmonic power can be kept unchanged with the reduction of V_{DD}?

As indicated in original Fig. S2, the parabolic I_D - V_G behaviour is slightly changing with V_{DS} . The fundamental reason is the band-to-band tunnelling electron transport mechanism governing this behaviour. As a result, two major changes occur when V_{DS} is changed (as original Fig. S2 shows):

1. The V_{peak} position is dependent on V_{DS} (as shown in Fig. 2). As we apply the input signal v_{in} with the same DC offset ($V_{offset} = -0.05$ V) for various V_{DS} in original Supplementary Information Fig. S7, the v_{out} waveform shape is slightly distorted as V_{peak} is not equal to V_{offset} . This slight waveform distortion leads to harmonics, thereby lowering the spectral purity (increasing the harmonic power);
2. The ideal parabolic I_D - V_G region narrows with decreasing V_{DD} (see figure below). Therefore, if we keep the same amplitude of v_{in} , a part of the input signal operates beyond the parabolic region, which also generate harmonic frequencies.

Based on the above explanation, in principle, the harmonic power cannot be kept unchanged with only reducing V_{DD} . However, one can improve the spectral purity (suppressing the harmonic power) by

1. adjusting the v_{in} DC-offset so that $V_{offset} = V_{peak}$ for each V_{DS}
2. adjusting the amplitude of v_{in} to ensure that it is smaller than the ideal parabolic region of operation.

To clarify this, we have added the figure below into Supplementary Fig. 8 in the revised Supplementary Information.

Supplementary Fig. 8. Frequency doubling at low V_{DD}e, The I_D - V_G curve in the low- V_{peak} state at V_{DD} of 0.5 V and 0.3 V with corresponding fitting, showing decreased parabolic operation region when decreasing V_{DD} .

We also explained the reason of this in Supplementary Fig. 8 caption in the revised Supplementary Information highlighted in yellow as well as below:

Notably, the output power at the desired doubled frequency is decreased from 98% to 88% when reducing V_{DD} from 0.5 V to 0.3 V. One reason is a slight output waveform distortion caused by the V_{peak} shift with changing V_{DD} , which generates additional harmonics as the DC-offset of v_{in} is not correspondingly adjusted. Another reason is that the high amplitude part of the input signal is operating beyond the parabolic I_D - V_G region due to a narrower current peak at lower V_{DD} . By tuning the v_{in} DC-offset and reducing its amplitude in accordance with the specific V_{DD} , the concentration of output power at doubled frequency is expected to increase.

We accordingly added explanation in lines 225-229 on page 10 in the revised manuscript highlighted in yellow as well as below:

As a result of the parabolic transfer curve due to the robust NTC properties (Supplementary Fig. 3), spectral purity remains high when further reducing the drive voltage, reaching 88% at $V_{DD} = 0.3$ V and 83% at ultra-low $V_{DD} = 50$ mV (Supplementary Fig. 8). The slight reduction in spectral purity at lower V_{DD} is due to the nonlinear change in the I_D - V_G characteristic with V_{DD} (detailed discussion in Supplementary Fig. 8).

5. The circuit architecture used in the manuscript seems completely the same as the previous works. The authors are suggested to add more data or discussion to highlight the difference.

We indeed acknowledge that we are using the perhaps simplest possible circuit architecture, i.e., a common-drain amplifier, also known as a source follower. The fact that we can reconfigurably achieve a multitude of signal modulation in such a simple circuit architecture is to us the true highlight of the manuscript. The simple circuit used in this work also simplifies the evaluation of the device performance in different modulation schemes. The authors in the previous works may have the same concern and thus a similar circuit with a very simple architecture was used. Of course, a more advanced circuitry may achieve even better performance, however, it is out of the scope of this proof-of-concept work.

We have clarified this in “Electrical characterizations” of “Methods” part in lines 326-328 on page 15 of the revised manuscript highlighted in yellow as well as below:

The waveforms of v_{out} were collected by the Rohde and Schwarz RTO Digital Oscilloscope with a conventional common-drain circuit configuration as shown in Fig. 3a.

6. The detailed measurement processes are missing. For example, the authors did not provide the method for realizing signal mixing in Figure 4. May the possible crosstalk between two input signals result in degradation of test results? If not, authors need to explain this. The method of power spectrum data was not described. It is not clear whether the data was obtained directly from oscilloscope or software analysis. If software analysis was used, the number of test cycles should be specified.

For the signal mixing measurement, we used a function generator to generate two sinusoidal signals with different frequencies (in our case, 800 Hz and 1 kHz) from two channels and then summed these two signals by using a T-shaped adapter (2-in 1-out). The output signal from the T-shaped adapter was connected to the

input of the device (gate terminal) as well as the oscilloscope to display the input waveform. In this configuration, the power spectrum obtained by applying a fast Fourier transform (FFT) to the input signal (see original Supplementary Information Fig. S8) shows that the other harmonics have < -50 dBm while the desired frequencies (800 Hz and 1 kHz) have the power of -3 dBm. Therefore, we think that the crosstalk between two signals is negligible in the measurements presented in Fig. 4, and we do not take that into consideration as no evident degradation is observed.

We have thus provided this fact in the figure caption of Supplementary Fig. 9 of the revised Supplementary Information highlighted in yellow as well as below:

The crosstalk between two input signals is negligible as the power spectrum (applying fast Fourier transform to entire v_{in}) of v_{in} shows a power of -3 dBm on the desired input signals (800 Hz and 1 kHz) while only < -50 dBm on other harmonics.

In light of the reviewer’s comment, we have now added a more detailed method of signal mixing measurement in “Electrical characterizations” of “Methods” part in lines 329-333 on page 15 of the revised manuscript highlighted in yellow as well as below:

For the signal mixing measurement presented in Fig. 4, a Rigol DG1000Z function generator was used to create two sinusoidal signals with different frequencies from two channels which were summed by using a BNC T-adaptor (2-in 1-out). The output signal from the T-adaptor is v_{in} used in the signal mixing measurements. The crosstalk between two input signals is negligible (detailed discussion in Supplementary Fig. 9).

We obtained the power spectra by using software analysis on the signal waveforms directly collected from the oscilloscope, as we described in the figure caption of Fig. 3 in the original manuscript. And we fully agree that the number of test cycles is critical to this as it defines the resolution of frequency-domain signal after applying FFT. In our case, we captured 100 ms signal from oscilloscope (as seen below) which corresponds to 100 cycles for the 1 kHz signal. The waveforms presented in the manuscript and the Supplementary Information is just a small representative portion of the measured waveforms.

We have now provided a more detailed method with the number of test cycles in “Electrical characterizations” of “Methods” part in lines 333-337 on page 15 of the revised manuscript highlighted in yellow as well as below:

All power spectra of v_{out} were obtained by applying the fast Fourier transform (FFT) algorithm on the collected waveforms directly from the oscilloscope with filtering of the DC component. To achieve high enough resolution of frequency-domain power spectra, 100 periods of all waveforms were captured from the oscilloscope.

We also clarified the corresponding description in the figure captions of Fig. 2 and Fig. 3 on page 22-23 of the revised manuscript highlighted in yellow as well as below:

Fig. 2:

c, Representative **excerpt of the** time-domain waveforms of v_{in} (a sinusoidal wave with $f_{in} = 1$ kHz) and i_{out} . The same result is obtained after 10-cycle reconfigurations. This can be used for BFSK to encode data as '1' and '0' in communication systems. *d*, The working principle for reconfigurable phase shift in the ferro-TFET. *e*, The demonstration of **the excerpt of the** time-domain i_{out} - v_{in} for reconfigurable phase shift in ferro-TFETs.

Fig. 3:

b. Representative **excerpt of the** time-domain waveforms of v_{in} and v_{out} in two reconfigurable states.

7. The hysteresis characteristic in the measured field-effect transfer curve is one of the important criteria to evaluate the quality of ferroelectric gate FET devices, which was displayed in the manuscript.

We fully agree with the reviewer that this is required for any implementation of a ferroelectric FET as a memory element (arguably, the by far most common usage of the technology). Therefore, even as the main application is not to use our device as a memory cell, we by convention show several instances of its memory capabilities by showing the transfer curve in the original manuscript and Supplementary Information such as Fig. 1g-h, and original Fig. S5a. As seen in original Fig. S6, we have a rather normal memory characteristic (but instead of using V_T , we use the for this manuscript more relevant property V_{peak}).

To clarify this, we added the corresponding description in lines 111-113 on page 5 of the revised manuscript highlighted in yellow as well as below:

*Transfer characteristics measured for the two distinct polarization states **with evident ferroelectric hysteresis** at various V_{DS} are plotted in Fig. 1h, confirming the reconfigurability of our ferro-TFETs.*

8. What is the theoretical limit of the proposed device, which is one of my concern in considering its promising application in signal modulation. Authors are suggested to further reduce the voltage value and pulse width of the voltage pulse to operate the device state, and explore the limit.

When considering its promising application in signal modulation, we have given more details about the theoretical frequency limit in our answer as a response to the initial general comment and to your **comment number 3**.

Regarding other limits, we do not see why an optimized structure would not be able to switch at voltages and pulse widths in correspondence with other ferroelectric FET implementations. Therefore, the theoretical limit on state switching should be in accordance with the voltage amplitude/pulse width limit of ferroelectric switching in the 13-nm-thick HZO. Further exploration of the minimum pulse width down towards the theoretical value of HZO ferroelectric switching speed (sub-nanosecond [7]) is currently limited by the DC probes in our measurement setup rather than by the device itself. Due to the nucleation limited switching of ferroelectric HfO₂, there is an inherent theoretical trade-off on the voltages and pulse widths required to reconfigure the device. One cannot reduce both at the same time without changing the device structure. By thinning the oxide, the coercive voltage decreases but at the same time there are indications that thinner films intrinsically switch slower than thicker films (still in the ns time regime) [8]. However, this structural optimization is beyond the scope of this work.

We have now clarified this in lines 104-109 on page 5 of the revised manuscript highlighted in yellow as well as below:

The ferroelectricity in the HZO gate oxide enables reconfigurable NTC in the ferro-TFETs. Depending on the V_{pulse} applied to the gate (+4 V/250 ns or -4 V/250 ns), the polarization in the ferroelectric film can be switched as shown in Fig. 1f, corresponding to the low- V_{peak} or high- V_{peak} state, respectively (inset of Fig. 1g; V_{peak} is defined by V_G at I_D peak). Although the amplitude of V_{pulse} can be substantially lowered by increasing the pulse width, ± 4 V / 250 ns has been optimized to balance the required voltage amplitude and pulse width.

9. The yield and uniformity of the electrical characteristic are critical to evaluate the promise of the novel device, and therefore authors should provide the related experimental data.

We thank the reviewer for this comment and have now provided statistics of pristine transfer characteristics (before any ferroelectric switching) from 20 devices on the same sample in Supplementary Fig. 2b of the revised Supplementary Information as well as below:

As the above figure shows, all devices exhibit NTC behaviour with average peak-to-valley-current-ratio (PVCR) above 10. The inset shows the statistical distribution of V_{peak} position of all measured devices, indicating a median V_{peak} at $V_G \sim 0.05$ V and a variation of about ± 50 mV. We attribute the limited uniformity in both V_{peak} and I_D mainly to the result of processing in a university lab with limited resources.

There is nothing in the fundamental physics implying that the variability caused by the ferroelectricity in these devices would be inferior to other scaled ferroelectric transistor implementations.

We clarify this in the Supplementary Fig. 2 caption of the revised Supplementary Information highlighted in yellow as well as below:

***b.* Transfer characteristics of 20 devices on the same sample. The inset shows the corresponding V_{peak} distribution, indicating a median V_{peak} sitting at $V_G \sim 0.05$ V and a variation of about ± 50 mV within 80% of the devices. The variation of V_{peak} and I_D among devices is mainly caused by processing variations but this does not change the conclusions of the work.**

also accordingly in lines 92-93 on page 5 of the revised manuscript highlighted in yellow as well as below:

Figure 1d shows the representative NTC ($g_m < 0$) with a peak-to-valley-current-ratio (PVCR, the I_D ratio between the peak point (3) and the valley point (4)) of about 50 in the transfer characteristic of the ferro-TFET with the measurement setup shown in Fig. 1c. Statistics of NTC behaviour are shown in Supplementary Fig. 2b.

References:

- [1] Science 332, 1294 (2011)
- [2] Nano Lett. 15, 1883 (2015)
- [3] IEEE Electron Device Lett. 30, 547 (2009)
- [4] Sci. Rep. 7, 46605 (2017)
- [5] Nat. Commun. 3, 1018 (2012)
- [6] ACS Nano 4, 10, 5532 (2010)

References from authors' response:

- [1] Ionescu AM, *et al.* Tunnel field-effect transistors as energy-efficient electronic switches. *Nature* **479**, 329-337 (2011).
- [2] Memisevic E, *et al.* Vertical InAs/GaAsSb/GaSb tunneling field-effect transistor on Si with $S = 48$ mV/decade and $I_{\text{on}} = 10 \mu\text{A}/\mu\text{m}$ for $I_{\text{off}} = 1 \text{ nA}/\mu\text{m}$ at $V_{\text{ds}} = 0.3 \text{ V}$. In: *2016 IEEE International Electron Devices Meeting (IEDM)* (2016).
- [3] Krishnaraja A, *et al.* Tuning of Source Material for InAs/InGaAsSb/GaSb Application-Specific Vertical Nanowire Tunnel FETs. *ACS Applied Electronic Materials* **2**, 2882-2887 (2020).
- [4] Memisevic E, *et al.* Impact of source doping on the performance of vertical InAs/InGaAsSb/GaSb nanowire tunneling field-effect transistors. *Nanotechnology*, 2018, 29(43): 435201.
- [5] Liu H, *et al.* Tunnel FET-based ultra-low power, low-noise amplifier design for bio-signal acquisition. *Proceedings of the 2014 international symposium on Low power electronics and design*. 2014: 57-62.
- [6] Lin Q, *et al.* Wearable Multiple Modality Bio-Signal Recording and Processing on Chip: A Review. *IEEE Sensors Journal* 21, 1108-1123 (2021).
- [7] Dahan M M, *et al.* Sub-Nanosecond Switching of Si: HfO₂ Ferroelectric Field-Effect Transistor. *Nano Letters*, 2023.
- [8] Lyu X, Si M, Shrestha P R, *et al.* First direct measurement of sub-nanosecond polarization switching in ferroelectric hafnium zirconium oxide. *IEEE International Electron Devices Meeting (IEDM)*. IEEE, 2019: 15.2. 1-15.2. 4.

To Reviewer #2:

The paper proposes to use ferroelectric TFET for reconfigurable analog applications. Leveraging the parabolic ID-VG characteristics of a ferro-TFET with gate/source overlapped channel, the authors demonstrated its application in frequency-doubling, 180 degree phase shift, signal transmission, and frequency mixing. The work is interesting and shows applications of ferroelectric devices in reconfigurable analog applications. The reviewer has a few recommendations.

1. The authors used the negative transconductance to achieve parabolic ID-VG characteristics. It is known that TFT shows ambipolar transport, which also features two branches. The authors are encouraged to further clarify the pros and cons of their design. The relevant reference of Fe-TFT are:

[1] J. Luo et al., "A Novel Ambipolar Ferroelectric Tunnel FinFET based Content Addressable Memory with Ultra-low Hardware Cost and High Energy Efficiency for Machine Learning," VLSI Symp. 2022

[2] J. Luo et al., "Novel Ferroelectric Tunnel FinFET based Encryption-embedded Computing-in-Memory for Secure AI with High Area-and Energy-Efficiency," IEDM 2022

We thank the reviewer for this question. We fully agree with the reviewer that the TFETs presented in the papers by Luo *et al.*, in principle, could be used for this purpose and there are actually several works cited in our manuscript that show similar function using ambipolar transistors [*refs. 9-11 in the original manuscript*]. To the best of our knowledge, the major benefit of using negative transconductance (NTC) TFET over ambipolar TFET is that the frequency doubling of NTC TFETs operates in the transistor on-state (at a current peak) whereas in ambipolar TFETs in the off-state (a current valley). The higher currents in the on-state results in a higher g_m , which enables a higher cut-off frequency for the NTC TFETs. This becomes especially important at low input voltages.

Furthermore, the transistor on-state for NTC TFETs leads to the second derivative of I_D to V_G being almost a constant (i.e., an ideal concave parabola) near the peak. As Fig. 1d shows, g_m is almost linear when transiting from the positive peak g_m to negative peak g_m , indicating that the NTC-based device can operate within this parabolic region for frequency doubling without generating harmonics. For ambipolar transistors, the current valley (convex shape transfer curve) is having an exponential I_D - V_G behaviour in the off-state leading to a non-ideal parabolic region for frequency doubling operation, which generates harmonics.

To clarify the comparison, we added more explanation with the above references that the reviewer listed in lines 54-62 on page 3 of the revised manuscript highlighted in yellow as well as below:

Advantageously, NTC in III-V heterostructure TFETs remains and even becomes stronger at lower source-drain bias (V_{DS})^{18,19}, allowing low-power analogue signal modulations such as phase shifting²¹. In the case of frequency doubling, compared to utilizing the current minimum in the symmetric transfer characteristic of ambipolar TFETs^{22,23}, NTC in TFETs allows frequency doubling around the maximum I_D in the on-state, which increases the operating frequency. Therefore, In fact, transistors with NTC have been recently under tremendous interests and intensively investigated in two-dimensional (2D) materials^{24,25} and organic heterojunctions^{26,27} with potentials for multi-valued logic gates^{24,26}, artificial synapses²⁵, and frequency doubling^{28,29}.

Also, in lines 217-219 on page 10 of the revised manuscript highlighted in yellow as well as below:

This differs from ambipolar TFETs^{22,23} which have exponential I_D - V_G dependence around the current valley in the subthreshold region. Due to this, NTC-based ferro-TFETs may double the frequency of a small input signal with a higher spectral purity at the output compared to ambipolar TFETs.

2. The authors are encouraged to show the switching speed of the device. In this work, only $\pm 4V$, 250ns write pulse is used. What is the switching speed of this device? With 13nm HZO, the hysteresis window of less than 0.5V is ultra-small. Could the authors comment on this?

We thank the reviewer for the suggestion. In fact, the voltage pulses used in this work ($\pm 4 V / 250 ns$) are actually rather optimized given our measurement setup and the current ferro-TFET structure. Further reduction in pulse width down towards the theoretical value of HZO ferroelectric switching speed (sub-nanosecond) [1] is currently limited by the DC probes in our measurement setup and seemingly not the TFET structure itself. We deem optimization of this as out of the scope of this work which is demonstrate the concept of reconfigurable signal modulation in a single device, but we discuss the question theoretically below.

Due to the nucleation limited switching of ferroelectric HfO_2 , there is an inherent theoretical trade-off on the voltages and pulse widths required to reconfigure the device. One cannot reduce both at the same time without changing the device structure. By thinning the oxide, the coercive voltage decreases but at the same time there are indications that thinner films intrinsically switch slower than thicker films [2]. In principle, we do not see why an optimized structure would not be able to switch at voltages and pulse widths in correspondence with other ferroelectric FET implementations (i.e., sub-nanosecond). Therefore, the theoretical limit should be in accordance with the voltage amplitude/pulse width limit of ferroelectric switching in the 13-nm-thick HZO, rather than the device itself.

We have now clarified this in lines 104-109 on page 5 of the revised manuscript highlighted in yellow as well as below:

The ferroelectricity in the HZO gate oxide enables reconfigurable NTC in the ferro-TFETs. Depending on the V_{pulse} applied to the gate ($+4 V/250 ns$ or $-4 V/250 ns$), the polarization in the ferroelectric film can be switched as shown in Fig. 1f, corresponding to the low- V_{peak} or high- V_{peak} state, respectively (inset of Fig. 1g; V_{peak} is defined by V_G at I_D peak). Although the amplitude of V_{pulse} can be substantially lowered by increasing the pulse width, $\pm 4 V / 250 ns$ has been optimized to balance the required voltage amplitude and pulse width.

Regarding the hysteresis/memory window, it is normally defined by the V_G difference at the same current level in two states. Using this conventional definition, the memory window of the presented ferro-TFET in Fig. 1g is about 0.84 V which is small but not unreasonably low for a 13-nm HZO film. The somewhat low value in Fig. 1g-h is due to the device being measured after many switching cycles after which the memory window has started to shrink. As seen in Fig. S6 in the original Supplementary Information, the initial memory window (ΔV_{peak}) during the first cycles is about 1.2 V, which is a rather common value. Low endurance is commonly seen for implementations of ferroelectrics on III-V. By defect and interface optimization, it can be expected that this behaviour may be vastly improved in the future.

We have now clarified this in lines 117-120 on page 6 of the revised manuscript highlighted in yellow as well as below:

The difference of V_{peak} between the two states (ΔV_{peak}) slightly increases with V_{DS} and approaches 0.45 V at $V_{DS} = 0.5 V$ (Fig. 1h). This value is somewhat small compared to other FeFET implementations but is mainly a result of memory window degradation after many switching cycles.

3. On page 6, the authors mentioned the application of the device in frequency shift keying. Along with many applications in Fig.2 and Fig.3, the modulated output amplitudes are also changing. This is because the high- V_{peak} and low- V_{peak} has a different amplitude. How does the unequal amplitude impact the overall

performance of the applications? In addition, the authors mentioned to program the device as a way of encoding the digital data. This brings a significant challenge to device endurance. The authors need to clarify that.

The unequal amplitude affects the output voltage somewhat due to a larger current flowing through the device as well as the different current range in two states (see figure below). In turn this means that one state may work at a slightly higher frequency than the other. This is not ideal as we would prefer high currents in both states, but the unevenness is not a problem in itself for frequency modulation as the frequency component of the signal is more critical compared to its amplitude.

We have thus clarified this in lines 194-198 on page 9 of the revised manuscript highlighted in yellow as well as below:

Notably, the amplitude of v_{out} in the high- V_{peak} states is larger due to a wider operating current range in the I_D - V_G curve compared to that in the low- V_{peak} state when applying an input signal near the current peak in the low- V_{peak} state, which might lead to slightly different cut-off frequencies in two states. However, this unevenness will not affect the function of frequency modulation since the frequency component of v_{out} in two states is more critical in this application scheme.

Regarding the device endurance, we agree with the reviewer that it is a challenge to use the ferro-TFET to encode digital data by programming the device by two different polarization states. III-V ferroelectric integration is a nascent technology and has not yet developed to the levels achieved by Si, and thus a lower endurance is typically observed [3,4], but with defect and interface engineering there is no reason to believe that it will not be possible to achieve similar performance. Although further improvement of endurance is required to use two polarization states to encode massive data, in some special applications such as for the purpose of security, a short endurance may actually be preferred to disable the device once the data is transmitted.

We have clarified this limit of endurance for data encoding and its potential application for security (even though it has low endurance) in lines 181-187 on page 8-9 of the revised manuscript highlighted in yellow as well as below:

Moreover, we also measure the endurance of NTC. The measured device shows an ~~good~~ endurance of $>10^5$ pulsing cycles with stable V_{peak} value in the low- V_{peak} state (Supplementary Fig. 7), in line with other III-V ferroelectric integrations³² and early Si implementations⁴⁵. Although the measured endurance is on the low side thus making the proposed BFSK application challenging for our current

ferro-TFET, it may be still useful in some special applications such as security systems in which disabling functions of the device is beneficial to complicate the reverse-engineering.

Nevertheless, for current ferro-TFETs with good retention but yet to be improved endurance, the applications such as reconfigurable frequency doubling and phase shifting are currently more practical (as already indicated in the manuscript). The ferroelectric switching is then only intermittently performed resulting in a long lifetime of the device despite the limited endurance.

4. Fig.3b shows a smaller output amplitudes for low- V_{peak} than the high- V_{peak} , which seems contradictory to Fig.1g.

We thank the reviewer for this comment. The amplitude of the output signal depends on the current range rather than the peak current value in the transfer curve. As the figure shows below, although the peak current is higher in the low- V_{peak} state, the operating range of the current is smaller than that in the high- V_{peak} state due to the frequency doubling feature. Therefore, a smaller output amplitude for low- V_{peak} than the high- V_{peak} is observed in Fig. 3b which is not contradictory to Fig. 1g.

We have clarified this (as we did in the previous question) in lines 194-198 on page 9 of the revised manuscript highlighted in yellow as well as below:

Notably, the amplitude of v_{out} in the high- V_{peak} states is larger due to a wider operating current range in the I_D-V_G curve compared to that in the low- V_{peak} state when applying an input signal near the current peak in the low- V_{peak} state, which might lead to slightly different cut-off frequency in two states. However, this unevenness will not affect the function of frequency modulation since the frequency feature of v_{out} in two states is more critical in this application scheme.

5. The Fig.5f benchmarking is not entirely fair. It only reflects a partial picture. How about the operation frequency? Though the authors claim that similar device shows a cutoff frequency of 3GHz on page 7, it does not mean that the current device will has that performance given that the oxide capacitance is much thicker for ferroelectric. Even though the same cutoff frequency carries over for ferroelectric device, the speed of the device is still much smaller than its FeFET counterpart. For analog application benchmarking, the area may not be the only metric of interest.

We fully agree with the reviewer that the figure only captures a small part of the potential space of opportunity of metrics for benchmarking and certainly the area is not the only metric of interest. However, area also

determines the required current levels as it affects the capacitive contribution, and thus the power consumption. We believe that the current benchmarking accurately captures the most important highlights of our device: a single device implementation with a vertical nanowire geometry (significantly reduced size) with a reconfigurable operation at very low supply voltages.

We have thus strengthened the claim of area scaling regarding the benchmarking in lines 222-225 on page 10 of the revised manuscript highlighted in yellow as well as below:

...but very few can operate with a $V_{DD} < 1 V$ while retaining a scaled footprint per device below $1 \mu\text{m}^2$, thereby being challenging for low-power systems and high-density integration with other technologies on the same platform. Area scaling also lowers the power consumption as it reduces the intrinsic capacitive contribution of the device.

In fact, all benchmarked single-transistor frequency doublers from literature have an operating frequency below MHz (10 Hz to 200 kHz). Our ferro-TFETs have an intermediate operating frequency among all listed devices. It certainly does not have the highest possible operation frequency of all available technologies, but the highlight of the article (reconfigurable operation) is to the best of our knowledge not available in any other implementation with the same feature size or drive voltage. Also, if one takes IoT low-frequency applications (e.g., bio-signal sensing) into account, the size and V_{DD} are more important than a high cut-off frequency.

We therefore added the frequency range of the benchmarked devices in Fig. 3 caption in lines 549-552 on page 23 of the revised manuscript highlighted in yellow as well as below:

f, Benchmarking of this work against other single-transistor frequency doublers. The transistor area from literature is based on the product of channel width and length or the estimation from the top-view microscope image. The operating frequency reported in listed work ranges from 10 Hz to 200 kHz. N.A. denotes “not available”.

6. In the discussion, the authors claim that the V_{DD} can be as low as 50mV, which could be confusing. It just shows that the drain supply voltage can be low. The gate voltage is still much larger. In addition, the necessity of +4/-4V programming also brings significant overhead.

We thank the reviewer for this comment and yes, in principle, we just show that V_{DD} can be as low as 50 mV. This is very important for low-power applications as it means that one can trade operating frequency for power consumption if power is more important than speed in the specific application.

The gate voltage (or here the input amplitude), however, will not contribute much to the power consumption due to the high impedance of the gate oxide (assuming a lowered parasitic capacitance to lower dynamic power dissipation). Also, the input amplitude typically depends on the device providing the input signal and the fact that our device handles both high and small input voltages is beneficial (even if that may mean that the gate voltage is much larger than V_{DD}).

In our case, we chose the same input signal amplitude for various V_{DD} mainly for the purpose of a fair comparison. In principle, a smaller gate voltage could be used for our device. Also, as we discussed in Fig. 3e, a smaller gate voltage could be even better in terms of suppressing harmonics as such a small input signal will completely operate within the ideal parabolic I_D - V_G region.

We have clarified this in the Supplementary Fig. 8 caption of the revised Supplementary Information highlighted in yellow as well as below:

Time-domain v_{in} - v_{out} characteristics at (a) $V_{DD} = 0.3 V$ and (c) $V_{DD} = 0.05 V$ with $f_{in} = 1 \text{ kHz}$. For a fair comparison, the same amplitude (0.3 V) of the input signal is kept at various V_{DD} .

...

In a low-power system with some specific application schemes, a small v_{in} may apply to the device. In principle, such a small signal should be detectable with our ferro-TFET and would operate fully within the ideal parabolic I_D - V_G region at an ultra-low V_{DD} .

We agree with the reviewer that the programming voltage (+4/-4V) is larger than, for instance, the CMOS digital logic compatible voltages (1.2 V). However, this voltage can be lowered by increasing the pulse width. Further reduction of programming voltage could be also achieved by thinning the ferroelectric film to reach below 1.2 V [5] Thus, it is possible, and it should be implementable also in our device architecture to realize a low programming voltage compatible with CMOS digital logic. Nevertheless, the highlight of this work is mainly the proof-of-the-concept of a new reconfigurable device variant for signal modulations, and the optimization of the device structure for a lowered programming voltage is beyond the scope of the current manuscript.

Therefore, we commented this in lines 104-109 on page 5 of the revised manuscript highlighted in yellow as well as below:

*The ferroelectricity in the HZO gate oxide enables reconfigurable NTC in the ferro-TFETs. Depending on the V_{pulse} applied to the gate (+4 V/250 ns or -4 V/250 ns), the polarization in the ferroelectric film can be switched as shown in Fig. 1f, corresponding to the low- V_{peak} or high- V_{peak} state, respectively (inset of Fig. 1g; V_{peak} is defined by V_G at I_D peak). *Although the amplitude of V_{pulse} can be substantially lowered by increasing the pulse width, ± 4 V / 250 ns has been optimized to balance the required voltage amplitude and pulse width.**

References from authors' response:

- [1] Dahan M M, Mulaosmanovic H, Levit O, et al. Sub-Nanosecond Switching of Si: HfO₂ Ferroelectric Field-Effect Transistor. *Nano Letters*, 2023.
- [2] Lyu X, Si M, Shrestha P R, et al. First direct measurement of sub-nanosecond polarization switching in ferroelectric hafnium zirconium oxide. *IEEE International Electron Devices Meeting (IEDM)*. IEEE, 2019: 15.2. 1-15.2. 4.
- [3] Athle R, Blom T, Irish A, et al. Improved Endurance of Ferroelectric HfxZr1-xO2 Integrated on InAs Using Millisecond Annealing. *Advanced Materials Interfaces*, 2022, 9(27): 2201038.
- [4] Persson A E O, Athle R, Svensson J, et al. A method for estimating defects in ferroelectric thin film MOSCAPs. *Applied Physics Letters*, 2020, 117(24): 242902.
- [5] Dutta S, Ye H, Khandker A A, et al. Logic compatible high-performance ferroelectric transistor memory. *IEEE Electron Device Letters*, 2022, 43(3): 382-385.

To Reviewer #3:

In this paper, author designed a reconfigurable ferroelectric TFET. And the diverse functionalities of signal modulations including frequency transmission, doubling, mixing, and phase can be achieved in this single device. The context and topic in this paper is interesting, but the author needs to further revise the article for the following problems:

1. The author only analyzed the working principle and test results of the device, and needs to further analyze the influence of structure parameters on device performance.

We do agree that the structure parameters of both the nanowire TFET and the ferroelectric HZO film have influence on the device performance. In this work, we have chosen relatively standard and well-characterized TFET and HZO parameters in order to prove the concept of reconfigurable signal modulations in a single device. We deem experimentally optimizing the structure to be beyond the scope of this manuscript, but we will theoretically discuss it below.

Typically, the memory window of FeFETs is proportional to the oxide thickness and thus a smaller memory window can be achieved when thinning the film. The main reason for using HZO and the specific thickness (13 nm) is 1) the CMOS-compatibility of HfO₂-based ferroelectrics, 2) to obtain an easily distinguishable memory window and 3) the ease of achieving ferroelectricity by rapid thermal annealing at the relatively low temperature of 400~500 °C. Thinning the HZO would enable lower operating gate voltages for the ferroelectric switching as well as improving the electrostatic control of the transistor channel, but may require higher annealing temperature to crystalize the HZO film for ferroelectricity, which may affect the III-V material quality of our device.

We have thus added the relevant discussion about HZO structure parameters in lines 109-111 on page 5 of the revised manuscript highlighted in yellow as well as below:

Here, 13-nm-thick HZO is used as the gate oxide in our ferro-TFETs as such films exhibit robust ferroelectricity at a thermal budget of below 500 °C^{32,33}. This is beneficial for III-V materials which may lack thermal stability at higher annealing temperatures.

The influence of structure parameters on a III-V nanowire TFET has been comprehensively studied in our previous works. For instance, in ref. 17 of the original manuscript (also see ref. [1] below) we showed that tuning the composition of As in the source material would change the performance of the TFET in both on- and off-state. Sample D (identical heterostructure to this manuscript) was optimized with high As composition (about 90%) and showed a balanced on- and off-state performance. In ref. [2], we have investigated the impact of doping concentration in the source segment on the TFET and found that fully doped source segment with intermediate doping level showed the best balance of on- and off-state.

As TFETs are governed by a small volume around the staggered heterostructure, the devices can in principle be physically scaled considerably with retained performance. Our previous work has decreased the channel diameter down to 10 nm which improved the subthreshold swing down to 35 mV/dec [3].

For the current device structure with gate/source overlap, we can tune the tunnelling length at the tunnel junction by moving the band edges of not only the InAs channel but also the (In)GaAsSb source segment at high gate bias (see band diagram in Fig. 1e). This is the key for realization of this parabolic transfer characteristic in the on-state. In principle, the more gate-overlapping region in the source, the longer tunnelling length at high gate bias (more suppression of tunnelling), leading to stronger source depletion and lower current.

Based on the above discussion, we have now included more analysis regarding III-V TFET structure in lines 79-83 on page 4 of the revised manuscript highlighted in yellow as well as below:

The nanowire TFET structure consists of three main segments, n-type doped InAs as the drain, non-intentionally doped (nid) InAs as the channel, and p-type doped (In)GaAsSb/GaSb as the source (Fig. 1a). This TFET structure is chosen to balance the on- and off-state performance, which has been systematically investigated in previous works including tuning source materials¹⁷ and doping concentrations¹⁹ as well as device scalability³⁰.

Moreover, as we mentioned in the original manuscript and the Supplementary Information, the limiting factor for the operating frequency for the current device structure is the parasitic capacitance, which can be considerably decreased by inserting a low-*k* spacer such as hydrogen silsesquioxane (HSQ) or SiO₂ between the electrodes.

We have now explained more in lines 165-169 on page 8 of the revised manuscript highlighted in yellow as well as below:

An optimized process with low-permittivity spacers such as hydrogen silsesquioxane (HSQ) or SiO₂ can mitigate the parasitic capacitances in vertical nanowire transistors, which can extend the operating frequency to GHz range^{40,41} (detailed discussion in Supplementary Fig. 5). In fact, vertical III-V nanowire TFETs with similar device structure but with low-permittivity spacers has shown a cut-off frequency up to 3 GHz⁴².

2. The author should give detailed parameters of ferroelectric material HZO in the paper.

We thank the reviewer for this comment. As we discussed in the **comment #1** above, the structure parameters such as the thickness of ferroelectric HZO film has now been added also to the main body of the revised manuscript in lines 109-111 on page 5 highlighted in yellow as well as below:

Here, 13-nm-thick HZO is used as the gate oxide in our ferro-TFETs as such films exhibit robust ferroelectricity at a thermal budget of below 500 °C^{32,33}. This is beneficial for III-V materials which may lack thermal stability at higher annealing temperatures.

We also added more detail about how we confirm the HZO thickness in lines 311-312 on page 14 in the “Methods” part of the revised manuscript highlighted in yellow as well as below:

Next, 13-nm HZO was grown at 200 °C for the gate oxide by thermal atomic layer deposition with a 1:1 alternation between the precursors TDMAHf and TEMAZr using water as the oxygen source. The thickness of the film was confirmed by ellipsometry...

More structural details such as polycrystalline characteristic and interface quality of HZO on InAs have been confirmed by grazing incidence X-ray diffraction (GIXRD) and high-resolution transmission electron microscopy (HRTEM), respectively, in our previous study with identical thickness, deposition conditions, and similar annealing temperature of HZO films [4]. Therefore, we believe that the structure and material properties of HZO used in this work should be nearly identical as that provided in ref. [4].

We have thus added this information in lines 312-313 on page 14 in the “Methods” part of the revised manuscript highlighted in yellow as well as below:

...and structure details such as polycrystalline characteristic and interface quality of HZO on InAs can be found in previous work³².

3. The author mentioned in the paper that the peak current is lower in the high- V_{peak} state than that in the low- V_{peak} state. The exact cause of this is also analyzed.

We thank the reviewer for this comment, and yes, we have analysed this phenomenon in lines 105-109 on page 5 of the original manuscript:

...caused by the different impact from the gate polarization on the source and channel region, which may further alter the factors that determine the maximum tunnelling current, such as the height and width of the tunnel barrier, and the density-of-states in the source region. Nevertheless, the difference in maximum I_D will not change the conclusions of this work.

References from authors' response:

- [1] Krishnaraja A., Svensson J., et al. Tuning of Source Material for InAs/InGaAsSb/GaSb Application-Specific Vertical Nanowire Tunnel FETs. *ACS Applied Electronic Materials* **2**, 2882-2887 (2020).
- [2] Memisevic E, et al. Impact of source doping on the performance of vertical InAs/InGaAsSb/GaSb nanowire tunneling field-effect transistors. *Nanotechnology*, 2018, 29(43): 435201.
- [3] Memisevic E, Svensson J, Lind E, et al. Vertical nanowire TFETs with channel diameter down to 10 nm and point S MIN of 35 mV/decade. *IEEE Electron Device Letters*, 2018, 39(7): 1089-1091.
- [4] Persson A E O, Athle R, Littow P, et al. Reduced annealing temperature for ferroelectric HZO on InAs with enhanced polarization. *Applied Physics Letters*, 2020, 116(6): 062902.

REVIEWERS' COMMENTS

Reviewer #1 (Remarks to the Author):

My comments have been well addressed in the revised manuscript and i can recommend it for publication.

Reviewer #2 (Remarks to the Author):

Thanks the authors for addressing my comments. Appreciate the comment addressing the difference between the proposed work and the ambipolar TFET.

Reviewer #3 (Remarks to the Author):

After reviewed the revised version, it can be accepted to publish.

Response Letter to Reviewers

Reviewer #1 (Remarks to the Author):

My comments have been well addressed in the revised manuscript and I can recommend it for publication.

We thank the reviewer for the useful comments and suggestions during the review process to help improve this work.

Reviewer #2 (Remarks to the Author):

Thanks the authors for addressing my comments. Appreciate the comment addressing the difference between the proposed work and the ambipolar TFET.

We thank the reviewer for the useful comments and suggestions during the review process to help improve this work.

Reviewer #3 (Remarks to the Author):

After reviewed the revised version, it can be accepted to publish.

We thank the reviewer for the useful comments and suggestions during the review process to help improve this work.